# Neural Vector Lyapunov–Razumikhin Certificates for Delayed Interconnected Systems

Jingyuan Zhou [1]   Yuexuan Wang [2]   Kaidi Yang [1]

## Abstract

Ensuring scalable input-to-state stability (sISS) is critical for the safety and reliability of large-scale interconnected systems, especially in the presence of communication delays. While learning-based controllers can achieve strong empirical performance, their black-box nature makes it difficult to provide formal and scalable stability guarantees. To address this gap, we propose a framework to synthesize and verify neural vector Lyapunov-Razumikhin certificates for discrete-time delayed interconnected systems. Our contributions are three-fold. First, we establish a sufficient condition for discrete-time sISS via vector Lyapunov-Razumikhin functions, which enables certification for large-scale delayed interconnected systems. Second, we develop a scalable synthesis and verification framework that learns the neural certificates and verifies the certificates on reachability-constrained delay domains with scalability analysis. Third, we validate our approach on mixed-autonomy platoons, drone formations, and microgrids against multiple baselines, showing improved verification efficiency with competitive control performance.

## 1. Introduction

Large-scale interconnected systems arise in many real-world applications, including intelligent transportation (Zhou & Yang, 2024; Zhou et al., 2024; 2026), robotics (Ren et al., 2023), and smart grids (Silva et al., 2021; Milanović & Zhu, 2017). Advances in sensing and communication technologies have made these systems increasingly complex and tightly coupled. A central challenge in controlling such systems is to ensure stability of the overall closed-loop dynamics.

In practice, such interconnected systems operate under substantial uncertainties, e.g., modeling mismatch (Zhang et al., 2019), exogenous disturbances (Vafamand et al., 2021), time-varying operating conditions (Chen et al., 2021), and unreliable communication (Zhang et al., 2022; Wang & Su, 2022). Among these factors, delays introduced by sensing, computation, and networked transmission are particularly difficult to address and often dominate performance degradation, making delay-awareness a key requirement for stability analysis and controller design.

Control strategies for ensuring stability in delayed interconnected systems can be broadly categorized into model-based and learning-based approaches. Model-based controllers, such as barrier Lyapunov-based control (Zenati et al., 2023), model predictive control (Zhu & Fridman, 2020), and sliding mode control (Yang et al., 2023a), explicitly incorporate stability into control objectives and constraints. This is often achieved by modeling delays directly (Wang et al., 2025) or embedding delay-compensation mechanisms (Molnár et al., 2017) into the controller design. However, these approaches typically require accurate system models, and their effectiveness can degrade substantially when applied to delayed interconnected systems with unknown dynamics. To overcome these limitations, learning-based methods for modeling dynamics (Zhu et al., 2021; Schlaginhaufen et al., 2021; Holt et al., 2022; Ji & Orosz, 2024; Stephany et al., 2024) and controllers (Sun et al., 2019; Yuan et al., 2023; Zhang et al., 2023d; Chen et al., 2024) have gained increasing popularity due to their ability to manage unknown dynamics and delay effects. However, the black-box nature of neural networks makes it challenging to provide guarantees for stability in delayed systems. The only notable attempt is Hedesh & Siami (2025), which requires the closed-loop system to be internally positive. This restricts its applicability, since many practical dynamics are not positive. In most existing neural network-based controllers, stability is treated as a soft constraint (Liotet et al., 2022; Holt et al., 2023), without offering rigorous stability guarantees. Consequently, formally guaranteeing stability in large-scale delayed interconnected systems remains an

---

[1]Department of Civil and Environmental Engineering, National University of Singapore [2]Institute of Operations Research and Analytics, National University of Singapore. Correspondence to: Kaidi Yang <kaidi.yang@nus.edu.sg>.

*Proceedings of the 43rd International Conference on Machine Learning*, Seoul, South Korea. PMLR 306, 2026. Copyright 2026 by the author(s).

open and challenging problem.

*Statement of Contribution.* To bridge the research gap, this work develops a scalable delay-aware certification framework for large-scale interconnected systems by synthesizing and verifying neural Lyapunov-Razumikhin certificates in discrete time. Our contributions are threefold. First, we establish a sufficient condition for discrete-time scalable input-to-state stability (sISS) of delayed interconnected systems using vector Lyapunov-Razumikhin functions, enabling scalable stability certification of such systems. Second, we propose a scalable synthesis-and-verification pipeline that learns neural certificates and verifies them over reachability-constrained delay domains. To address scalability, we develop theoretical foundations that enable the reuse of certificates across structurally equivalent systems, thereby improving verification efficiency. Third, we demonstrate the effectiveness of the proposed method on mixed-autonomy platoons, drone formations, and microgrids, where it improves verification efficiency while achieving competitive control performance compared with multiple baselines.

## 2. Related Work

This work makes contributions in two main research areas: (i) stability of delayed dynamic systems, and (ii) neural certificates and stable-by-design neural control.

### 2.1. Stability of Delayed Dynamic Systems

Stability analysis for time-delayed systems has a long history, with classical tools including Lyapunov-Krasovskii functionals (Mazenc et al., 2012; Zhang et al., 2016; 2018; Aleksandrov, 2024) and Lyapunov-Razumikhin techniques (Pepe, 2017; Schlaginhaufen et al., 2021; Nekhoroshikh et al., 2022; Liu et al., 2023). However, most delay-dependent conditions are derived for fixed-size networks, limiting large-scale applicability. This contrasts with scalable stability results for delay-free interconnected systems (Qiu et al., 2024). To the best of our knowledge, explicitly scalable delay-aware stability conditions for general interconnected systems remain unavailable.

### 2.2. Neural Certificates and Stable-by-Design Neural Control

To provide guarantees for learning-based controllers, existing methods generally fall into two categories: certificate-based verification and stable-by-design neural control.

The first approach is neural certificate-based method, which learns Lyapunov- or barrier-type certificates (Dai et al., 2021; Clark, 2021; Qin et al., 2021; Yang et al., 2024; 2023b; Zhang et al., 2023b;c; Schlaginhaufen et al., 2021; Hu et al., 2024; Tayal et al., 2024; Zhang et al., 2025; Mandal et al., 2024a; Liu et al., 2024; Nadali et al., 2024; Zhou et al.,

2025; Neustroev et al., 2025; Yu et al., 2025; Wang et al., 2024). These certificates can be formally verified with control theoretical-based approaches (Anand & Zamani, 2023; Tayal et al., 2024; Zhang et al., 2024; Henzinger et al., 2025; Ren et al., 2025) and neural network verification tools like (Wang et al., 2021; Lopez et al., 2023; Wu et al., 2024). These methods allow for counterexample-guided inductive synthesis (Ding et al., 2022; Mandal et al., 2024b; Zhao et al., 2024) or certified training (Mueller et al., 2023), thereby synthesizing controllers that satisfy barrier or Lyapunov properties. Nonetheless, to the best of the authors' knowledge, no existing work has proposed neural certificates for large-scale delayed interconnected systems.

The second direction is stable-by-design neural control, which embeds stability or robustness guarantees directly into the neural architecture or controller parameterization. Examples include compositional port-Hamiltonian neural controllers (Furieri et al., 2022; Cheng et al., 2024; Zakwan & Ferrari-Trecate, 2026) and stable neural feedback parameterizations with robustness guarantees (Barbara et al., 2025; Manchester et al., 2026). However, the former relies on dissipativity-oriented structural assumptions and does not explicitly consider communication delays, while the latter mainly targets single nonlinear systems rather than delayed interconnected dynamics. Hence, these methods are not directly applicable to large-scale delayed interconnected systems.

## 3. Problem Statement

Consider an interconnected system of $N$ agents indexed by $i \in \mathcal{N} = \{1, \ldots, N\}$. The interconnection topology is captured by an adjacency matrix $G \in \{0, 1\}^{N \times N}$, where $G_{i,j} = 1$ indicates that the dynamics of agent $i$ depend on the state of agent $j$, and $G_{i,j} = 0$ otherwise. We represent the system as a tuple $\mathcal{I} = (\mathcal{N}, \{\mathcal{E}_i\}_{i \in \mathcal{N}}, \{f_i\}_{i \in \mathcal{N}})$, where $\mathcal{E}_i = \{j \in \mathcal{N} \mid G_{i,j} = 1\}$ denotes the neighbor set of agent $i$, and $f_i$ denotes its local dynamics:

$$x_{i,k+1} = f_i\Big(x_{i,k}, u_{i,k}, \{x_{j,k-s_{ij}}\}_{j \in \mathcal{E}_i}, d_{i,k}\Big), \quad (1)$$

where $x_{i,k} \in \mathbb{R}^{n_i}$ denotes the state of agent $i$ at time step $k$, $u_{i,k} \in \mathbb{R}^{p_i}$ is its control input, and $\{x_{j,k-s_{ij}}\}_{j \in \mathcal{E}_i}$ denotes the neighbor states received by agent $i$. The delay variable $s_{ij}$ is the communication delay from agent $j$ to agent $i$, bounded by a known constant $\tau_{i,j} \in \mathbb{Z}_{\geq 0}$. For notation simplicity, we define $\tau_{\max} = \max_{i \in \mathcal{N}, j \in \mathcal{E}_i} \tau_{i,j}$. The term $d_{i,k} \in \mathcal{W}_i \subset \mathbb{R}^{n_i}$ represents an exogenous disturbance acting on agent $i$ at time step $k$, where $\mathcal{W}_i$ is a given bounded set. Let $x_k := \{x_{i,k}\}_{i=1}^N \in \mathbb{R}^{\sum_{i=1}^N n_i}$ denote the stacked system state. Define $\tau_{\max} := \max_{i,j} \tau_{ij}$. The initial condition is given by a history sequence $x_{i,k} = \phi_i(k), k \in \{-\tau_{\max}, -\tau_{\max} + 1, \ldots, 0\}$, for each $i \in \{1, \ldots, N\}$, where $\phi_i : \{-\tau_{\max}, -\tau_{\max} + 1, \ldots, 0\} \to \mathbb{R}^{n_i}$ denotes

the initial history of subsystem $i$.

*Remark* 3.1. We assume that sensing and actuation on each agent are delay-free (i.e., no self-state measurement delay and no control execution delay), and only the communication of neighbor states $\{x_{j,k}\}_{j\in\mathcal{E}_i}$ is subject to bounded delays. This setting is standard in networked multi-agent control (Jia et al., 2019).

Our goal is to design a neural network controller without assuming explicit knowledge of the local dynamics $f_i(\cdot)$ since accurate dynamics are usually hard to identify in the real world. Specifically, for each agent $i$, we seek a policy

$$u_{i,k} = \pi_i\Big(x_{i,k},\ \{x_{j,k-s_{ij}}\}_{j\in\mathcal{E}_i}\Big), \qquad (2)$$

such that the resulting closed-loop delayed interconnected system in (1) is stable for all admissible delays $s_{ij} \in \{1,\ldots,\tau_{i,j}\}$ and bounded disturbances $d_{i,k}$.

Specifically, we adopt a scalable input-to-state stability (sISS) notion under bounded delay, which requires the state to be uniformly bounded for any network size $N$. This leads to the desired property that the closed-loop robustness does not degrade as the interconnected system grows.

**Definition 3.2** (Discrete-Time Scalable Input-to-State Stability Under Bounded Delay). The discrete-time interconnected system in Eq. (1) is said to be *sISS* if there exist functions $\beta \in \mathcal{KL}$ and $\mu \in \mathcal{K}$, independent of the network size $N$, such that for any $k \in \mathbb{Z}_{\geq 0}, i \in \mathcal{N}$, any initial history $\{x_{i,-s}\}_{s=0}^{\tau}$, and any bounded disturbance sequence $d_i$, the following inequality holds:

$$\max_{i\in\mathcal{N}} |x_{i,k}|_2 \leq \beta \left( \max_{i\in\mathcal{N}} \max_{s\in\{0,\ldots,\tau\}} |x_{i,-s}|_2,\ k \right)$$
$$+ \mu \left( \max_{i\in\mathcal{N}} \|d_i\|_{\mathcal{L}_\infty} \right). \qquad (3)$$

Typically, for systems without delays, scalable input-to-state stability can be certified using sISS vector Lyapunov functions, e.g., by verifying a network-size-independent small-gain condition (Silva et al., 2024, Theorem 1). However, for delayed systems, the closed-loop dynamics are no longer memoryless, since each agent evolves based on delayed neighbor states, which invalidates a direct application of the delay-free vector Lyapunov argument. This motivates delay-aware certificates that explicitly account for the bounded delay $\tau_{\max}$ while preserving network-size-independent stability constants.

## 4. Discrete-time sISS Vector Lyapunov Razumikhin Function Formulation

We provide a sufficient condition for sISS of delayed systems (see Definition 3.2) based on Lyapunov functions and a Razumikhin-type condition that handles the state delay. The condition is given in Theorem 4.1 (see Appendix A.1 for proof).

**Theorem 4.1** (Discrete-Time sISS via Vector Lyapunov-Razumikhin Functions). *The discrete-time delayed system in Eq.* (1) *is sISS if there exist scalars* $p > 1$, $\epsilon \in (0,1)$, $\psi \geq 0$, *class-*$\mathcal{K}_\infty$ *functions* $\alpha_1, \alpha_2 \in \mathcal{K}_\infty$, *Lyapunov functions* $V_i : \mathbb{R}^{n_i} \to \mathbb{R}_{\geq 0}$, $i \in \mathcal{N}$, *positive gains* $\Gamma = \{\gamma_{i,j}\}$ *satisfying the small-gain condition*

$$\max_{i\in\mathcal{N}} \sum_{j\in\mathcal{E}_i\cup\{i\}} \gamma_{i,j} \leq (1-\epsilon), \qquad (4)$$

*such that the following conditions hold:*

*(i) Class-*$\mathcal{K}_\infty$ *Bounds: For all* $i \in \mathcal{N}$ *and all states* $x_{i,k}$:

$$\alpha_1(|x_{i,k}|_2) \leq V_i(x_{i,k}) \leq \alpha_2(|x_{i,k}|_2) \qquad (5)$$

*(ii) Conditional Decrement: For all* $i \in \mathcal{N}$, *if the Razumikhin condition*

$$\max_{\substack{j\in\mathcal{E}_i\cup\{i\} \\ s\in\{1,\ldots,\tau_{\max}\}}} V_j(x_{j,k-s}) \leq p \cdot V_i(x_{i,k}) \qquad (6)$$

*holds at time* $k$, *then the following decremental inequality is satisfied:*

$$V_i(x_{i,k+1}) \leq \sum_{j\in\mathcal{E}_i\cup\{i\}} \gamma_{i,j} V_j(x_{j,k}) + \psi\|d_{i,k}\|_2 \quad (7)$$

*Remark* 4.2. Based on Theorem 4.1 and its proof, a valid choice of $\beta$ and $\mu$ in Definition 3.2 is:

$$\beta(r,k) = \alpha_1^{-1}\big(c\,\rho^k\alpha_2(r)\big), \qquad \mu(r) = \alpha_1^{-1}\Big(\frac{\psi}{\epsilon}r\Big), \quad (8)$$

where $\rho = \max\left\{\exp^{-\frac{\ln p}{\tau_{\max}+1}},\ 1-\epsilon\right\} \in (0,1), c \geq p$

The sufficient condition in Theorem 4.1 provides a practical way to verify the discrete-time sISS property for the delayed interconnected system in Eq. (1). Specifically, it requires constructing a family of local Lyapunov functions $\{V_i\}_{i\in\mathcal{N}}$ that satisfy the class-$\mathcal{K}_\infty$ bounds and the Razumikhin-type conditional decrement in Theorem 4.1. In the decrement inequality, the coupling coefficient $\gamma_{i,j}$ quantifies the influence of subsystem $j$ on subsystem $i$, while the small-gain constraint enforces a network-size-independent contraction. However, these coupled constraints and the delay-dependent Razumikhin condition make the manual design of $\{V_i\}$ challenging. Hence, we next propose a learning-based approach to synthesize and verify such Lyapunov-Razumikhin certificates.

# 5. Synthesis and Verification Framework

In this section, we present the proposed framework for formal synthesis and verification of discrete-time sISS certificates in delayed interconnected systems. Section 5.1 introduces the synthesis module, which trains the neural certificates and controllers. Section 5.2 formulates verification as a global logical condition over reachability-constrained delay domains and adopts a two-stage verification strategy over a prescribed level set. Section 5.3 presents scalable algorithms that leverage structural properties of specific classes of interconnected systems to keep training and verification tractable.

## 5.1. Synthesis of Neural Certificates

We synthesize neural sISS certificates by jointly searching for the coupling matrix and Lyapunov functions satisfying conditions in Theorem 4.1. To this end, we parameterize the coupling matrix $\Gamma$ by combining a learnable matrix $\Gamma_{\mathrm{pure}}$ and adjacency matrix $G$ as $\Gamma = \mathcal{Q}\big(\operatorname{ReLU}(\Gamma_{\mathrm{pure}}) \circ G\big)$, where the elementwise product $\circ$ ensures $\gamma_{i,j} = 0$ when $G_{i,j} = 0$, and the ReLU activation guarantees $\gamma_{i,j} \geq 0$. Function $\mathcal{Q}$ ensures the small-gain condition by converting any $\gamma_{i,j}$ to $\min\left\{1, \frac{1-\varepsilon}{\sum_{j \in \mathcal{E}_i \cup \{i\}} \gamma_{i,j}}\right\} \gamma_{i,j}$, $j \in \mathcal{E}_i \cup \{i\}$. The matrix $\Gamma_{\mathrm{pure}}$ is then trained together with Lyapunov functions.

To enforce Eqs. (5)-(7), we jointly train Lyapunov functions and $\Gamma$ as:

$$L_{\mathrm{total}}(\Gamma, V) = w_{\mathrm{imi}} L_{\mathrm{imi}} + w_p L_p + w_d L_d, \tag{9}$$

$$L_{\mathrm{imi}} = \frac{1}{|\mathcal{N}|} \sum_{i \in \mathcal{N}} \|\pi_i - \pi_{i,\mathrm{nom}}\|_2 \tag{10}$$

$$L_p = \frac{1}{|\mathcal{N}|} \sum_{i \in \mathcal{N}} \Big( \operatorname{ReLU}(\alpha_1(|x_{i,k}|_2) - V_i(x_{i,k}) + \epsilon_p)$$
$$+ \operatorname{ReLU}(-\alpha_2(|x_{i,k}|_2) + V_i(x_{i,k}) + \epsilon_p) \Big), \tag{11}$$

$$L_d = \frac{1}{|\mathcal{N}|} \sum_{i \in \mathcal{N}} \operatorname{ReLU}\big(-\max(\mathrm{con}_{\neg A}, \mathrm{con}_B) + \epsilon_d\big), \tag{12}$$

where $w_{\mathrm{imi}}, w_p, w_d > 0$ are weighting factors. The first term $L_{\mathrm{imi}}$ is to keep the learned controller close to a well-performing nominal policy $\pi_{i,\mathrm{nom}}$. The second term $L_p$ enforces the Class-$\mathcal{K}_\infty$ bounds, with $\alpha_1$ and $\alpha_2$ been chosen as linear functions with positive slopes and the margin $\epsilon_p \geq 0$. The third term $L_d$ enforces the Razumikhin conditional decrement condition, with

$$\mathrm{con}_{\neg A} = \max_{\substack{j \in \mathcal{E}_i \cup \{i\} \\ s \in \{1, \ldots, \tau_{\max}\}}} V_j(x_{j,k-s}) - p \cdot V_i(x_{i,k}), \tag{13}$$

$$\mathrm{con}_B = \sum_{j \in \mathcal{E}_i \cup \{i\}} \gamma_{i,j} V_j(x_{j,k}) + \psi\|d_{i,k}\|_2 - V_i(x_{i,k+1}), \tag{14}$$

where $\epsilon_d \geq 0$ is a margin. This reformulation is valid since $A \Rightarrow B \iff \neg A \vee B$.

*Remark* 5.1. To construct the training set, we sample initial states from a prescribed initial state space and roll out a fixed number of trajectories using the nominal controller, each with the same horizon length. For time indices prior to the sampled initial state (i.e., the delayed history required by $\tau_{\max}$), we pad the trajectory by setting the unavailable past states to the equilibrium state.

## 5.2. Verification Formulation

This subsection formalizes the verification task for a candidate neural certificate $(\Gamma, V)$ learned in Section 5.1. Given the learned Lyapunov-Razumikhin functions $\{V_i\}_{i \in \mathcal{N}}$ and coupling gains $\Gamma = \{\gamma_{i,j}\}$, we aim to certify that the Razumikhin conditional logic in Theorem 4.1 holds for all admissible delayed state histories within a prescribed region of interest.

First, we give the logical condition corresponding to the Razumikhin conditional decrement in Theorem 4.1. Specifically, for each agent, at least one of the following two expressions must hold:

$$\bigwedge_{i \in \mathcal{N}} \left( \left( \max_{\substack{j \in \mathcal{E}_i \cup \{i\} \\ s \in \{1, \ldots, \tau_{\max}\}}} V_j(x_{j,k-s}) > p \cdot V_i(x_{i,k}) \right) \vee \right.$$
$$\left. \left( V_i(x_{i,k+1}) \leq \sum_{j \in \mathcal{E}_i \cup \{i\}} \gamma_{i,j} V_j(x_{j,k}) + \psi\|d_{i,k}\|_2 \right) \right) \tag{15}$$

where $\vee$ denotes the logical OR operator, $\wedge$ denotes the logical AND operator. For ease of verification, we rewrite the implication form of the Razumikhin logic as an equivalent disjunction using $A \Rightarrow B \iff \neg A \vee B$. Together with the class-$\mathcal{K}_\infty$ bounds in Eq. (5), this logical condition forms the certificate requirements in Theorem 4.1.

Direct verification over the entire state space is intractable. To address this, we restrict attention to delay histories that are reachable from a given initial delayed-state set $S_0$ over a finite horizon. Here $S_k \subset \mathbb{R}^{(\tau_{\max}+1)\sum_{i \in \mathcal{N}} n_i}$ denotes an over-approximation of the reachable set of the global delay-history state $\mathrm{col}(x_k, \ldots, x_{k-\tau_{\max}})$ at time step $k$. Let $\mathrm{Reach}(f, S)$ denote a one-step reachability operator for the corresponding one-step (delay-embedded) dynamics of $\mathrm{col}(x_k, \ldots, x_{k-\tau_{\max}})$. Given $S_0$ and a horizon $T \in \mathbb{N}$, define recursively the over-approximated reachable sets for $k = 1, \ldots, T$ by

$$S_k := \mathrm{Reach}(f, S_{k-1}). \tag{16}$$

For each $k \in \{0, \ldots, T\}$ and each $i \in \mathcal{N}$, define the

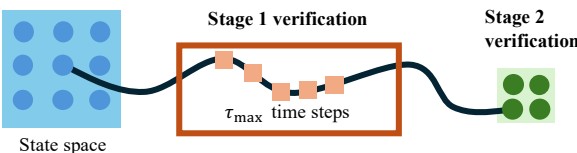

*Figure 1.* Two-stage verification framework.

reachability-constrained delay domain

$$\mathcal{Z}_{i,k}^* := \Big\{ \{\{x_{j,k-s}\}_{s=0}^{\tau_{\max}}\}_{j \in \mathcal{E}_i \cup \{i\}} : x_{j,k-s} \in \mathrm{Proj}_{j,s}(S_k),$$
$$\forall j \in \mathcal{E}_i \cup \{i\},\ s = 0, \dots, \tau_{\max} \Big\}, \qquad (17)$$

where $\mathrm{Proj}_{j,s}(\cdot)$ denotes the projection of a global delay-history set onto the state coordinates of agent $j$ at lag $s$. This construction yields a verification domain that is both system-consistent and horizon-dependent.

*Remark* 5.2. The reachability operator $\mathrm{Reach}(\cdot)$ in (16) can be instantiated using sampling-based methods, which are suitable when the closed-loop dynamics are a black box. A broad class of approaches generates rollouts from the initial set and constructs an outer approximation of the reachable set with theoretical guarantees such as Ramdani et al. (2009); Liebenwein et al. (2018); Zhang et al. (2023a).

To further improve tractability, we adopt a two-stage strategy (as in Fig. 1) that separates verification into an "outside" region, where the Razumikhin conditional logic is enforced on reachable delay histories, and an "inside" region, where a strengthened unconditional contraction is verified on a target level set. This leads to an sISS sufficient condition written in Theorem 5.3.

**Theorem 5.3** (Two-stage verification). *Fix* $R \geq \frac{\psi}{\epsilon}\bar{d}$, *where* $\bar{d} := \sup_{k \geq 0} \max_{i \in \mathcal{N}} \|d_{i,k}\|_2$. *Let* $T_R := \left\lceil \frac{\ln\left(R/(cV_{\max}(0))\right)}{\ln \rho} \right\rceil$, *where* $c \geq p$ *and* $\rho \in (0,1)$ *are the constants given by Remark 4.2, and* $V_{\max}(k) = \max_{i \in \mathcal{N}} V_i(x_{i,k})$. *Assume that: (i) for all $k$ with $V_{\max}(k) > R$, the Razumikhin-based conditional logic is verified on the reachability-constrained domain $\mathcal{Z}_{i,k}^*$; (ii) for all states with $V_{\max} \leq R$, Eqs. (5) and (7) are verified on the entire set for agent $i$ as*

$$\mathcal{S}_{R,i} := \mathrm{Proj}_{\mathcal{E}_i \cup \{i\}}(\{x : V_{\max}(x) \leq R\}), \qquad (18)$$

*where* $\mathrm{Proj}_{\mathcal{E}_i \cup \{i\}}(\cdot)$ *denotes the projection of a global set onto the stacked state coordinates* $\mathrm{col}(\{x_j\}_{j \in \mathcal{E}_i \cup \{i\}})$. *Then, any trajectory with* $\mathrm{col}(x_0, \dots, x_{-\tau_{\max}}) \in S_0$ *enters* $\{V_{\max} \leq R\}$ *by time* $T_R$ *and satisfies* $V_{\max}(k) \leq R$ *for all* $k \geq T_R$.

The detailed proof of Theorem 5.3 is given in Appendix A.2.

Finally, we derive a sampling-based sufficient condition for Theorem 5.3 to make verification tractable. Specifically,

we reduce the universal verification requirements to finitely many checks while retaining correctness for all continuous states. To achieve this, we introduce the following assumptions for (i) the Lipschitz continuity of $f_i$ and $V_i$ and (ii) a grid-based sampling.

**Assumption 5.4** (Lipschitz Continuity). For each agent $i \in \mathcal{N}$, the system dynamics $f_i$ and the Lyapunov-Razumikhin functions $V_i$ are Lipschitz continuous with constants $L_{f_i}$ and $L_{V_i}$, respectively.

**Definition 5.5** (Uniform Grid Sampling of Local Delay-History Domains). Fix an agent $i \in \mathcal{N}$ and a time index $k$. Let the local delay-history variable

$$z_{i,k} := \Big( \{x_{j,k-s}\}_{j \in \mathcal{E}_i \cup \{i\},\, s=0,\dots,\tau_{\max}},\, d_{i,k} \Big) \in \mathbb{R}^{m_i^{\mathrm{out}}} \tag{19}$$

lie in a bounded region of interest $\mathcal{Z}_{i,k}^* \subset \mathbb{R}^{m_i^{\mathrm{out}}}$, where $m_i^{\mathrm{out}} := (\tau_{\max} + 1)\sum_{j \in \mathcal{E}_i \cup \{i\}} n_j + n_d$ and $n_d$ is the disturbance dimension.

Construct a finite dataset $\mathcal{D}_{i,k} \subset \mathcal{Z}_{i,k}^*$ by uniform gridding: for each coordinate $q = 1, \dots, m_i^{\mathrm{out}}$, discretize the $q$-th coordinate interval of $\mathcal{Z}_{i,k}^*$ with step size $\Delta_{i,k}^{(q)} > 0$. Denote $\Delta_{i,k} := (\Delta_{i,k}^{(1)}, \dots, \Delta_{i,k}^{(m_i^{\mathrm{out}})})$.

For any $z_{i,k} \in \mathcal{Z}_{i,k}^*$, let $\hat{z}_{i,k} = \Pi_{\mathcal{D}_{i,k}}(z_{i,k})$ be a nearest grid point in $\mathcal{D}_{i,k}$. Then

$$\|z_{i,k} - \hat{z}_{i,k}\|_2 \leq \tfrac{1}{2}\|\Delta_{i,k}\|_2. \tag{20}$$

**Definition 5.6** (Uniform Grid Sampling on $\mathcal{S}_{R,i}$). Fix an agent $i \in \mathcal{N}$ and define $z_i := (\{x_j\}_{j \in \mathcal{E}_i \cup \{i\}}, d_i) \in \mathbb{R}^{m_i^{\mathrm{in}}}$ with $m_i^{\mathrm{in}} := \sum_{j \in \mathcal{E}_i \cup \{i\}} n_j + n_d$. Construct a uniform grid dataset $\mathcal{D}_i^{\mathrm{in}} \subset \mathcal{S}_{R,i}$ with per-coordinate step sizes $\Delta_i^{\mathrm{in}} = (\Delta_i^{(1)}, \dots, \Delta_i^{(m_i^{\mathrm{in}})})$. For any $z_i \in \mathcal{S}_{R,i}$, let $\hat{z}_i \in \mathcal{D}_i^{\mathrm{in}}$ (for notational simplicity, we drop $k$ when the domain is time-invariant) be a nearest grid point. Then

$$\|z_i - \hat{z}_i\|_2 \leq \tfrac{1}{2}\|\Delta_i^{\mathrm{in}}\|_2. \tag{21}$$

*Remark* 5.7. Assumption 5.4 and Definitions 5.5-5.6 are mild in our setting and hold for a broad class of smooth physical systems on compact operating domains (Nejati et al., 2023). We next clarify how the associated constants are obtained in practice.

(i) For each agent $i \in \mathcal{N}$, the Lipschitz constant $L_{f_i}$ in Assumption 5.4 can be derived a priori from the analytic model on the compact operating set, e.g., by upper bounding the Jacobian norm of $f_i$, and is treated as fixed. The constants $L_{V_i}$ are estimated offline using neural Lipschitz bounding methods such as (Xu & Sivaranjani, 2024), which provide sound upper bounds for feed-forward networks over a prescribed domain.

(ii) For Definitions 5.5 and 5.6, the grid resolution $\Delta_{i,k}, \Delta_i^{\text{in}}$ controls the covering granularity of the sampled dataset $\mathcal{D}_{i,k}, \mathcal{D}_i^{\text{in}}$. For example, for any $z_{i,k} \in \mathcal{Z}_{i,k}$ there exists a grid point $\hat{z}_{i,k} \in \mathcal{D}_{i,k}$ whose component-wise distance to $z_{i,k}$ is bounded by $\Delta_{i,k}/2$. Combined with the Lipschitz constants in Assumption 5.4, this yields a deterministic margin that transfers the sampled inequalities to all continuous points in $\mathcal{Z}_{i,k}^*$.

Moreover, the class-$\mathcal{K}_\infty$ bounds condition (5) only depends on the state of a single agent without delay, and can therefore be verified directly over the continuous domain using neural network verification tools such as Marabou (Wu et al., 2024). Under the above assumptions, the infinite-dimensional verification problem over continuous delay histories can be reduced to finite checks on sampled points with appropriate margins. This yields the following Theorem.

**Theorem 5.8** (Formal Verification of Vector Lyapunov-Razumikhin Conditions)**.** *Consider the delayed interconnected system under Assumption 5.4 and Definitions 5.5-5.6 and the class-$\mathcal{K}_\infty$ condition (5) is satisfied. Fix $R \geq \frac{\psi}{\epsilon} \bar{d}$ with $\bar{d} := \sup_{k \geq 0} \max_{i \in \mathcal{N}} \|d_{i,k}\|_2$. For each agent $i \in \mathcal{N}$ and time index $k$, let $\mathcal{Z}_{i,k}^*$ be the reachability-constrained local delay-history domain and define the inside set $\mathcal{S}_{R,i}$ as in Theorem 5.3. Let $\mathcal{D}_i^{\text{out}} \subset \mathcal{Z}_{i,k}^* \setminus \mathcal{S}_{R,i}$ and $\mathcal{D}_i^{\text{in}} \subset \mathcal{S}_{R,i}$ be uniform-grid datasets. Let $\varepsilon_{i,k}^{\text{out}} := \frac{1}{2}\|\Delta_{i,k}\|_2$, $\varepsilon_{i,k}^{\text{in}} := \frac{1}{2}\|\Delta_i^{\text{in}}\|_2$.*

*Suppose that the following sampled conditions hold:*

*Stage 1: outside. For all sampled points $\hat{z}_{i,k} \in \mathcal{D}_{i,k}^{\text{out}}$,*

$$
\begin{pmatrix}
\left( pV_i(x_{i,k}) - \max_{\substack{j \in \mathcal{E}_i \cup \{i\} \\ s \in \{1,\ldots,\tau_{\max}\}}} V_j(x_{j,k-s}) < -L_{h_i,upper}\varepsilon_{i,k}^{out} \right) \vee \\
\left( V_i(x_{i,k+1}) \leq \sum_{j \in \mathcal{E}_i \cup \{i\}} \gamma_{ij} V_j(x_{j,k}) + \psi\|d_{i,k}\|_2 - \delta_{i,k}^{out} \right)
\end{pmatrix},
$$
(22)

*and the margin satisfies $\delta_{i,k}^{\text{out}} \geq L_{r_i,upper}\varepsilon_{i,k}^{out}$, where $L_{r_i,upper} = L_{V_i} L_{f_i} + \sum_{j \in \mathcal{E}_i \cup \{i\}} |\gamma_{ij}| L_{V_j} + \psi$, $L_{h_i,upper} = p\, L_{V_i} + \max_{j \in \mathcal{E}_i \cup \{i\}} L_{V_j}$.*

*Stage 2: inside. For all sampled points $\hat{z}_i \in \mathcal{D}_i^{\text{in}}$,*

$$
V_i(x_{i,k+1}) \leq \sum_{j \in \mathcal{E}_i \cup \{i\}} \gamma_{ij} V_j(x_{j,k}) + \psi\|d_{i,k}\|_2 - \delta_i^{\text{in}},
$$
(23)

*and the margin satisfies $\delta_i^{\text{in}} \geq L_{r_i,upper}\epsilon_i^{\text{in}}$.*

*Then, the two-stage delayed-sISS conditions in Theorem 5.3 hold on the corresponding continuous domains. Hence, the trajectory reaches $\{V_{\max} \leq R\}$ in finite time and stays there thereafter.*

The detailed proof of Theorem 5.8 is given in Appendix A.3. If a candidate certificate $(\Gamma, \{V_i\}_{i \in \mathcal{N}})$ fails the above verification, we can refine it via a counterexample-guided inductive synthesis (CEGIS) loop. Specifically, the verifier returns violating samples from the reachable delay domains, which are then added to the training set to re-synthesize $(\Gamma, \{V_i\}_{i \in \mathcal{N}})$. This CEGIS loop is repeated until the certificate is verified. Algorithmic details are provided in Appendix B.

### 5.3. Scalability Analysis

The proposed framework for synthesizing neural certificates can be computationally challenging for large-scale interconnected systems. To address this, we develop a scalable synthesis framework that accelerates certificate construction by reusing certificates learned for smaller or structurally equivalent systems.

Theorem 5.10 shows that, under the structural conditions in Definition 5.9, sISS certificates for a larger delayed interconnected system can be obtained by reusing certificates from a smaller system. Intuitively, this requires that every substructure of $\tilde{\mathcal{I}}$ is embedded in $\mathcal{I}$ via the injective map.

**Definition 5.9** (Substructure Isomorphism for Delayed Systems)**.** A system with delays $\tilde{\mathcal{I}} = (\tilde{\mathcal{N}}, \{\tilde{\mathcal{E}}_j\}_{j \in \tilde{\mathcal{N}}}, \{\tilde{f}_j\}_{j \in \tilde{\mathcal{N}}})$ is substructure-isomorphic to $\mathcal{I} = (\mathcal{N}, \{\mathcal{E}_i\}_{i \in \mathcal{N}}, \{f_i\}_{i \in \mathcal{N}})$, if, for each $j \in \tilde{\mathcal{N}}$, there exists a map $\iota_j : \{j\} \cup \tilde{\mathcal{E}}_j \to \mathcal{N}$ satisfying: $\tilde{f}_j = f_{\iota_j(j)}, \iota_j(\tilde{\mathcal{E}}_j) = \mathcal{E}_{\iota_j(j)}$, and, for each $l \in \tilde{\mathcal{E}}_j$, the delay $\tilde{s}_{jl}$ is identical to the corresponding delay, i.e., $\tilde{s}_{jl} = s_{\iota_j(j)\iota_j(l)}$.

**Theorem 5.10** (sISS Preservation under Substructure Isomorphism for Delayed Systems)**.** *Suppose an interconnected system with delays $\mathcal{I}$ admits a vector Lyapunov–Razumikhin function $\{V_i\}_{i \in \mathcal{N}}$ as its sISS certificate. Suppose $\tilde{\mathcal{I}}$ is substructure-isomorphic to $\mathcal{I}$ in the sense of Definition 5.9. Further suppose that, for any $j, j' \in \tilde{\mathcal{N}}$ and any $l \in (\tilde{\mathcal{E}}_j \cup \{j\}) \cap (\tilde{\mathcal{E}}_{j'} \cup \{j'\})$, the corresponding certificates are consistent, i.e., $V_{\iota_j(l)} = V_{\iota_{j'}(l)}$. Then, $\tilde{\mathcal{I}}$ admits vector Lyapunov–Razumikhin certificates $\{\tilde{V}_j\}_{j \in \tilde{\mathcal{N}}}$, where $\tilde{V}_j = V_{\iota_j(j)}$ for all $j \in \tilde{\mathcal{N}}$. Moreover, the corresponding coupling gains can be chosen as $\tilde{\gamma}_{jl} = \gamma_{\iota_j(j)\iota_j(l)}$ for all $j \in \tilde{\mathcal{N}}$ and $l \in \tilde{\mathcal{E}}_j \cup \{j\}$.*

The proof of Theorem 5.10 is given in Appendix A.4. Then, definition 5.11 extends this notion to the node level by formalizing structural equivalence between agents (see Appendix C for an illustrative example). Theorem 5.12 then shows that structurally equivalent nodes can share an identical certificate, which can substantially reduce verification complexity.

**Definition 5.11** (Node Structural Equivalence for Delayed Systems)**.** Let $\mathcal{I}$ be an interconnected system with delays.

A node $j \in \mathcal{N}$ is said to be structurally equivalent to node $\iota(j)$ under a permutation $\iota$ of $\mathcal{N}$ if for each $j \in \mathcal{N}$: (1) $f_j = f_{\iota(j)}$, (2) $\iota(\mathcal{E}_j) = \mathcal{E}_{\iota(j)}$, and (3) for each $l \in \mathcal{E}_j$, the delay $s_{jl}$ is identical to the corresponding delay $s_{\iota(j)\iota(l)}$.

**Theorem 5.12** (Identical Certificates for Structurally Equivalent Nodes in Delayed Systems). *Suppose an interconnected system with delays $\mathcal{I}$ satisfies the sISS conditions. Then, there exists a sISS certificate $\{V_i\}_{i \in \mathcal{N}}$ such that $V_i = V_j$ for any structurally equivalent nodes $i, j \in \mathcal{N}$.*

Theorem 5.12 implies that $\mathcal{N}$ can be partitioned into structural equivalence classes as in Definition 5.11, with all nodes in the same class sharing an identical certificate. The proof is given in Appendix A.5. Consequently, verification can be simplified by checking only a single representative node from each class.

# 6. Numerical Simulation

In this section, we conduct numerical simulations to evaluate the proposed framework. Section 6.1 evaluates the verification efficiency of the proposed method. Section 6.2 compares the control performance with benchmark methods. We evaluate our proposed method in three application environments, including vehicle platoon (Zhou et al., 2026), drone formation control (Ghamry & Zhang, 2015), and microgrid (Silva et al., 2021). Section 6.3 presents the simulation visualizations. Detailed experiment settings are given in Appendix D.

## 6.1. Verification Performance

Table 1 compares four verification designs under two delay horizons $\tau_{\max} \in \{1, 5\}$: (i) *Ours*, which combines *reachability-constrained delay domains* as in Theorem 5.8 with a *scalable sISS verification structure* as in Theorems 5.10 and 5.12; (ii) *Reachability only*, which only uses the reachability-constrained domains verification design; (iii) *Scalability only*, which keeps the scalable structure but verifies on the original delay domains as in Eq. (15); and (iv) *Direct*, which performs verification directly on the original domains without either design. 'TO' indicates timeout when the verification runtime exceeds 5 hours.

The results show that combining reachability reduction and structural reuse is crucial for practical runtimes in large-scale delayed systems. Ours is the only design that consistently finishes across all scenarios and scales, while its runtime remains nearly invariant as $N$ increases. This indicates that the verification cost is dominated by local subproblems rather than the global system dimension. In contrast, Reachability only can complete for small-scale instances ($N{=}10$) but is about one order of magnitude slower than Ours and times out for $N{=}50$, showing that shrinking the domain alone cannot prevent the dimension blow-up. Moreover,

Scalability only and Direct time out in all cases, implying that without reachability-constrained domains the verification remains overly conservative and computationally prohibitive even with a scalable structure.

## 6.2. Control Performance

To evaluate stability in the proposed delay-aware framework, we focus on the boundedness of the trajectory tracking error under delayed observations. Define the tracking error $e_{i,k} = x_{i,k} - x_i^\star$ with respect to the equilibrium $x_i^\star$. We measure the root-mean-square tracking error (RMSE) averaged over the horizon and all agents:

$$\text{RMSE} = \sqrt{\frac{1}{NT} \sum_{k=1}^{T} \sum_{i=1}^{N} \|e_{i,k}\|_2^2}, \qquad (24)$$

where $N$ is the number of agents in the network and $T$ is the total number of discrete time steps in the evaluation horizon. This RMSE serves as an empirical evidence for stability in the delay environment. We test varying disturbances into the system and report the resulting RMSE across the following benchmark methods:

**1. Nominal controller**: The original controller without stability certification. Reinforcement learning for platoon control (Zhou et al., 2026), LQR for UAV control (Ghamry & Zhang, 2015), and linear feedback control for microgrids (Silva et al., 2021).

**2. Predictor feedback** (Xu et al., 2022): A model-based delay-compensation baseline that predicts the current state and applies to the nominal controller.

**3. Compositional ISS** (Zhang et al., 2023c): A certificate baseline enforcing an ISS small-gain condition.

**4. sISS** (Qiu et al., 2024; Zhou et al., 2025): A certificate baseline targeting scalable ISS.

**5. Proposed method**: The delay-aware neural Lyapunov-Razumikhin certification framework.

Tables 2-4 summarize the RMSE under different disturbance settings in three environments. Across all three tasks, the proposed delay-aware method consistently achieves comparable results, indicating stronger disturbance attenuation and improved tracking performance.

In the platoon environment (Table 2), increasing the sinusoidal disturbance amplitude from 4 m/s to 7 m/s substantially increases RMSE for all methods. The proposed method achieves the lowest RMSE in both cases, reducing RMSE by 22.86% at 4 m/s and by 20.18% at 7 m/s, and further improving upon Compositional ISS by 5.81% and 1.66%, respectively. Notably, it also outperforms the predictor-feedback baseline, demonstrating comparable performance without an explicit vehicle model.

*Table 1.* Verification runtime comparison under different verification designs.

| Scenario | $N$ | Ours | | Reachability only | | Scalability only | | Direct | |
|---|---|---|---|---|---|---|---|---|---|
| | | $\tau_{\max}=1$ | $\tau_{\max}=5$ | $\tau_{\max}=1$ | $\tau_{\max}=5$ | $\tau_{\max}=1$ | $\tau_{\max}=5$ | $\tau_{\max}=1$ | $\tau_{\max}=5$ |
| Platoon | 10 | 385±2.9 | 372±7.4 | 4753± 22.6 | 4635±12.3 | TO | TO | TO | TO |
| Platoon | 50 | 377±1.3 | 380±2.4 | TO | TO | TO | TO | TO | TO |
| Drone | 10 | 1114±33.2 | 1243±6.3 | 11295±7.5 | 10868±6.2 | TO | TO | TO | TO |
| Drone | 50 | 1044±20.0 | 1106±16.4 | TO | TO | TO | TO | TO | TO |
| Microgrid | 6 | 1998±13.1 | 1970±10.1 | 8931±18.6 | 8853±9.5 | TO | TO | TO | TO |
| Microgrid | 50 | 1999±4.6 | 1961±11.2 | TO | TO | TO | TO | TO | TO |

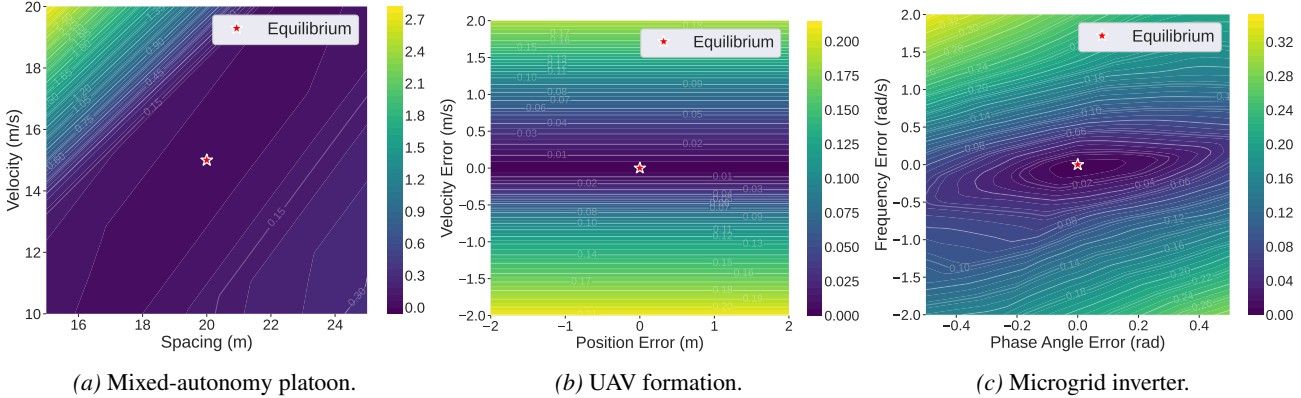

*(a)* Mixed-autonomy platoon.    *(b)* UAV formation.    *(c)* Microgrid inverter.

*Figure 2.* Lyapunov function evolution on three benchmarks under the proposed delayed-sISS framework.

*Table 2.* RMSE under sinusoidal disturbances for platoon environment.

| | 1/15 Hz, 4 m/s | 1/15 Hz, 7 m/s |
|---|---|---|
| Predictor Feedback | 1.04 | 2.12 |
| Nominal Controller | 1.05 | 2.23 |
| Compositional ISS | 0.86 | 1.81 |
| sISS | 0.95 | 1.97 |
| Proposed Method | **0.81** | **1.78** |

*Table 3.* RMSE under sinusoidal disturbances for drone formation environment.

| | 1/15 Hz, 0.5 m/s | 1/15 Hz, 3 m/s |
|---|---|---|
| Predictor Feedback | 34.211 | 34.317 |
| Nominal Controller | 34.584 | 34.712 |
| Compositional ISS | 34.574 | 34.704 |
| sISS | 34.583 | 34.710 |
| Proposed Method | **34.568** | **34.698** |

For drone formation (Table 3), RMSE values are numerically very close across controllers and disturbance levels. Relative to the nominal controller, the proposed method improves RMSE by only 0.046% and 0.040%, and by about 0.017% relative to Compositional ISS in both cases. It also matches the model-based predictor-feedback baseline, suggesting robust behavior under the tested settings.

In the microgrid environment (Table 4), RMSE increases with the initial disturbance magnitude as expected. The proposed method yields the lowest RMSE, reducing RMSE by 12.20% at 0.5 rad/s and by 2.92% at 1 rad/s, while also improving upon Compositional ISS by 8.86% and 6.34%, respectively. It also outperforms the model-based predictor-feedback baseline in both cases, achieving comparable performance without an explicit microgrid model.

### 6.3. Simulation Demonstration

**Lyapunov Visualization.** Figure 2 plots the neural Lyapunov-Razumikhin functions on three benchmarks. In all cases, the Lyapunov value is small near the equilibrium and increases as the state deviates, indicating a consistent energy-like measure.

**Trajectory Visualization.** Figure 3 shows the corresponding closed-loop trajectories. The platoon velocities converge to the desired value, the UAV velocities remain bounded around the desired velocity, and the inverter frequencies stay bounded and approach the nominal frequency.

### 7. Conclusion

In this paper, we developed a scalable delay-aware certification framework for interconnected systems. We established

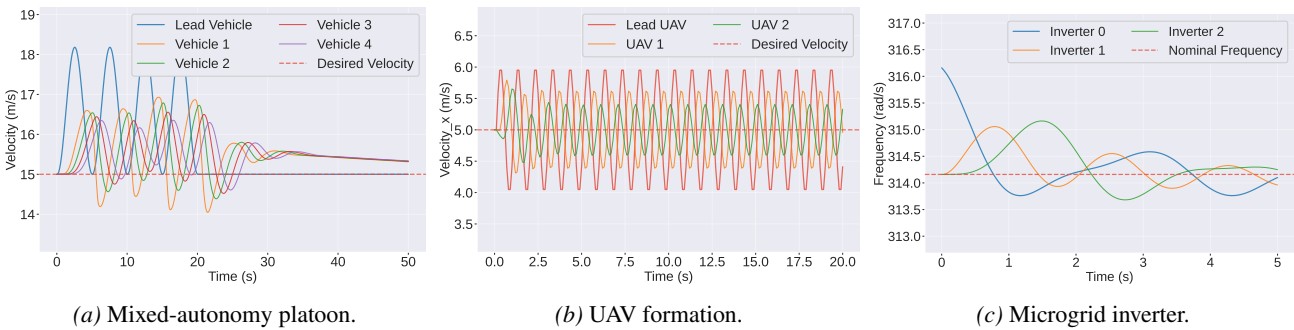

*(a)* Mixed-autonomy platoon.       *(b)* UAV formation.       *(c)* Microgrid inverter.

*Figure 3.* System trajectories on three benchmarks under the proposed delayed-sISS framework.

*Table 4.* RMSE under initial disturbances for microgrid environment.

|                      | 0.5 rad/s | 1 rad/s |
|----------------------|-----------|---------|
| Predictor Feedback   | 0.082     | 0.138   |
| Nominal Controller   | 0.082     | 0.137   |
| Compositional ISS    | 0.079     | 0.142   |
| sISS                 | 0.119     | 0.171   |
| Proposed Method      | **0.072** | **0.133** |

a sufficient condition for discrete-time sISS using vector Lyapunov-Razumikhin functions, and proposed a scalable synthesis-and-verification pipeline that learns neural certificates and verifies them over reachability-constrained delay domains. Across mixed-autonomy platoons, drone formations, and microgrids, the proposed approach improves verification efficiency while achieving competitive tracking performance in the presence of communication delays.

## Impact Statement

Our work advances machine learning methods for synthesizing and formally verifying stability certificates, and it can be broadly useful across cyber-physical systems where delays are unavoidable, including transportation, robotics, and power systems. Beyond these examples, the same framework may apply to other large-scale networked control settings. Overall, we do not anticipate impacts that require special discussion beyond this general applicability.

## Acknowledgement

This research was supported by the Singapore Ministry of Education (MOE) under its Academic Research Fund Tier 1 (A-8003262-00-00).

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

# A. Theoretical Results

## A.1. Proof of Theorem 4.1

*Proof.* Define the composite Lyapunov function

$$V_{\max}(k) := \max_{i \in \mathcal{N}} V_i(x_{i,k}). \tag{25}$$

By the class-$\mathcal{K}_\infty$ bounds in (5), there exist $\alpha_1, \alpha_2 \in \mathcal{K}_\infty$ such that

$$\alpha_1\left(\max_{i \in \mathcal{N}} |x_{i,k}|_2\right) \leq V_{\max}(k) \leq \alpha_2\left(\max_{i \in \mathcal{N}} |x_{i,k}|_2\right). \tag{26}$$

Thus, it suffices to establish an sISS bound for $V_{\max}(k)$.

**Case 1: Razumikhin condition not satisfied.** Suppose that at time $k$, for some agent $i$,

$$\max_{j \in \mathcal{E}_i \cup \{i\}} \max_{s \in \{1,\dots,\tau_{\max}\}} V_j(x_{j,k-s}) > p V_i(x_{i,k}), \qquad p > 1. \tag{27}$$

Then we have

$$V_i(x_{i,k}) \leq \tfrac{1}{p} \max_{j \in \mathcal{E}_i \cup \{i\}} \max_{s \in \{1,\dots,\tau_{\max}\}} V_j(x_{j,k-s}). \tag{28}$$

Taking the maximum over all $i$, which implies

$$V_{\max}(k) \leq \tfrac{1}{p} \max_{s \in \{1,\dots,\tau_{\max}\}} V_{\max}(k-s). \tag{29}$$

Applying this argument recursively, every $(\tau_{\max} + 1)$ steps $V_{\max}$ decreases by at least a factor $1/p$, which implies

$$V_{\max}(k) \leq p^{-\left\lfloor \frac{k}{\tau_{\max}+1} \right\rfloor} V_{\max}(0) \leq p * p^{-\frac{k}{\tau_{\max}+1}} V_{\max}(0) = p \exp\left(-\frac{\ln p}{\tau_{\max}+1} k\right) V_{\max}(0). \tag{30}$$

**Case 2: Razumikhin condition satisfied.** If condition Eq. (6) holds for agent $i$ at time $k$, then by Eq. (7),

$$V_i(x_{i,k+1}) \leq \sum_{j \in \mathcal{E}_i \cup \{i\}} \gamma_{i,j} V_j(x_{j,k}) + \psi \|d_{i,k}\|_2. \tag{31}$$

Taking the maximum over $i \in \mathcal{N}$ and using the small-gain condition (4),

$$V_{\max}(k+1) \leq (1-\epsilon) V_{\max}(k) + \psi \max_{i \in \mathcal{N}} \|d_{i,k}\|_2. \tag{32}$$

We expand this recursion step by step:

$$V_{\max}(1) \leq (1-\epsilon) V_{\max}(0) + \psi \max_i \|d_{i,0}\|_2, \tag{33}$$

$$V_{\max}(2) \leq (1-\epsilon) V_{\max}(1) + \psi \max_i \|d_{i,1}\|_2$$
$$\leq (1-\epsilon)^2 V_{\max}(0) + (1-\epsilon)\psi \max_i \|d_{i,0}\|_2 + \psi \max_i \|d_{i,1}\|_2, \tag{34}$$

$$V_{\max}(3) \leq (1-\epsilon)^3 V_{\max}(0) + (1-\epsilon)^2 \psi \max_i \|d_{i,0}\|_2 + (1-\epsilon)\psi \max_i \|d_{i,1}\|_2 + \psi \max_i \|d_{i,2}\|_2. \tag{35}$$

Continuing this expansion, we obtain the general bound

$$V_{\max}(k) \leq (1-\epsilon)^k V_{\max}(0) + \sum_{t=0}^{k-1} (1-\epsilon)^{k-1-t} \psi \max_{i \in \mathcal{N}} \|d_{i,t}\|_2. \tag{36}$$

The summation can be upper-bounded using the supremum of the disturbances and the finite geometric series:

$$\sum_{t=0}^{k-1}(1-\epsilon)^{k-1-t}\,\psi\max_i\|d_{i,t}\|_2 \le \psi\left(\sup_{t\le k-1}\max_i\|d_{i,t}\|_2\right)\sum_{t=0}^{k-1}(1-\epsilon)^t \tag{37}$$

$$\le \frac{\psi}{\epsilon}\sup_{t\le k}\max_i\|d_{i,t}\|_2. \tag{38}$$

Therefore, for all $k \ge 0$,

$$V_{\max}(k) \le (1-\epsilon)^k V_{\max}(0) + \frac{\psi}{\epsilon}\sup_{t\le k}\max_{i\in\mathcal{N}}\|d_{i,t}\|_2. \tag{39}$$

**Case 3: Switching between conditions.** At each time instant, either Case 1 (Razumikhin condition not satisfied) or Case 2 (Razumikhin condition satisfied) must hold. Thus, the evolution of $V_{\max}(k)$ is bounded by either the exponential decay in (30) or the geometric decay with disturbance input in (39). By the discrete-time comparison principle, if the system trajectory is governed at each step by one of two inequalities, then the overall trajectory can be bounded by the worst-case decay rate of the two. Hence, combining (30) and (39), we conclude that for all $k \ge 0$,

$$V_{\max}(k) \le c\,\rho^k V_{\max}(0) + \frac{\psi}{\epsilon}\sup_{t\le k}\max_{i\in\mathcal{N}}\|d_{i,t}\|_2, \tag{40}$$

where

$$\rho := \max\left\{\exp^{-\frac{\ln p}{\tau_{\max}+1}},\ 1-\epsilon\right\} \in (0,1), \qquad c \ge p \tag{41}$$

**Conclusion.** From the class-$\mathcal{K}_\infty$ bounds in (5), there exist functions $\beta \in \mathcal{KL}$ and $\mu \in \mathcal{K}$ defined by

$$\beta(r,k) := \alpha_1^{-1}\left(c\rho^k\alpha_2(r)\right), \qquad \mu(r) := \alpha_1^{-1}\left(\tfrac{\psi}{\epsilon}r\right), \tag{42}$$

such that the following bound holds for all $k \ge 0$:

$$\max_{i\in\mathcal{N}}|x_{i,k}|_2 \le \beta\left(\max_{i\in\mathcal{N}}|x_{i,0}|_2,\ k\right) + \mu\left(\sup_{t\le k}\max_{i\in\mathcal{N}}\|d_{i,t}\|_2\right)$$

$$\le \beta\left(\max_{i\in\mathcal{N}}|x_{i,0}|_2,\ k\right) + \mu\left(\max_{i\in\mathcal{N}}\|d_i\|_{\mathcal{L}_\infty}\right). \tag{43}$$

Moreover, since the initial history satisfies

$$\max_{i\in\mathcal{N}}\max_{s\in\{0,\ldots,\tau_{\max}\}}|x_{i,-s}|_2 \ge \max_{i\in\mathcal{N}}|x_{i,0}|_2, \tag{44}$$

we have:

$$\max_{i\in\mathcal{N}}|x_{i,k}|_2 \le \beta\left(\max_{i\in\mathcal{N}}\max_{s\in\{0,\ldots,\tau_{\max}\}}|x_{i,-s}|_2,\ k\right) + \mu\left(\max_{i\in\mathcal{N}}\|d_i\|_{\mathcal{L}_\infty}\right). \tag{45}$$

then the above inequality implies the definition of sISS in Eq. (3). This completes the proof. $\qquad\square$

## A.2. Proof of Theorem 5.3

*Proof.* Consider the composite Lyapunov level $V_{\max}(k) := \max_{i\in\mathcal{N}} V_i(x_{i,k})$. We analyze two regions separated by the threshold $R$.

**Outside region** ($V_{\max}(k) > R$) By the definition of the reachable sets and the domain $\mathcal{Z}_{i,k}^*$, the true trajectory satisfies $(x_{i,k},\ldots,x_{i,k-\tau_{\max}}) \in \mathcal{Z}_{i,k}^*$ for all relevant $k$ prior to leaving the reachability envelope. Hence, for every time instant such that $V_{\max}(k) > R$, hypothesis (i) guarantees that the original Razumikhin-based conditional logic holds along the true

trajectory at time $k$. Therefore, we invoke the conclusion of Theorem 4.1 on those time instants, i.e., there exist constants $c \geq p$ and $\rho \in (0, 1)$ such that the associated comparison bound holds:

$$V_{\max}(k) \leq c\,\rho^k V_{\max}(0). \tag{46}$$

Consequently, the time when the trajectory must have entered the level set $\{V_{\max} \leq R\}$ can be upper-bounded by solving

$$c\,\rho^T V_{\max}(0) \leq R \iff \rho^T \leq \frac{R}{cV_{\max}(0)} \tag{47}$$

$$\iff \ln(\rho^T) \leq \ln\Big(\frac{R}{cV_{\max}(0)}\Big) \tag{48}$$

$$\iff T \ln \rho \leq \ln\Big(\frac{R}{cV_{\max}(0)}\Big) \tag{49}$$

Since $\ln \rho < 0$, this is equivalent to

$$T \geq \frac{\ln\big(R/(cV_{\max}(0))\big)}{\ln \rho}. \tag{50}$$

Thus, choosing

$$T_R := \left\lceil \frac{\ln\big(R/(cV_{\max}(0))\big)}{\ln \rho} \right\rceil \tag{51}$$

guarantees $V_{\max}(T_R) \leq R$, i.e., the trajectory enters $\{V_{\max} \leq R\}$ no later than time $T_R$.

**Inside region** ($V_{\max}(k) \leq R$). By hypothesis (ii), on the entire set $\{V_{\max} \leq R\}$, the sISS condition as in Eq. (39) holds:

$$V_{\max}(k+1) \leq (1-\epsilon)V_{\max}(k) + \psi \max_{i \in \mathcal{N}} \|d_{i,k}\|_2. \tag{52}$$

Since the local radius is chosen such that

$$R \geq \frac{\psi}{\epsilon}\,\bar{d}, \qquad \bar{d} := \sup_{k \geq 0} \max_{i \in \mathcal{N}} \|d_{i,k}\|_2, \tag{53}$$

for any $k$ with $V_{\max}(k) \leq R$, we have

$$V_{\max}(k+1) \leq (1-\epsilon)R + \psi\bar{d} \tag{54}$$

$$\leq (1-\epsilon)R + \epsilon R \tag{55}$$

$$\leq R, \tag{56}$$

which shows that the set $\{V_{\max} \leq R\}$ is forward invariant.

Combining the two parts, the trajectory enters $\{V_{\max} \leq R\}$ by time $T_R$ and, by forward invariance, satisfies $V_{\max}(k) \leq R$ for all $k \geq T_R$. $\qquad \square$

**A.3. Proof of Theorem 5.8**

*Proof.* Fix $R \geq \frac{\psi}{\epsilon}\,\bar{d}$, where $\bar{d} := \sup_{k\geq 0} \max_{i \in \mathcal{N}} \|d_{i,k}\|_2$, and let $T_R = \left\lceil \frac{\ln\big(R/(cV_{\max}(0))\big)}{\ln \rho} \right\rceil$, where $c \geq p$ and $\rho \in (0,1)$ are the constants in Remark 4.2. Fix an agent $i \in \mathcal{N}$ and a time index $k$.

Define the Razumikhin residual and decrement residual as functions of $z_{i,k}$:

$$h_i(z_{i,k}) := p\,V_i(x_{i,k}) - \max_{\substack{j \in \mathcal{E}_i \cup \{i\} \\ s \in \{1,\dots,\tau_{\max}\}}} V_j(x_{j,k-s}), \tag{57}$$

$$r_i(z_{i,k}) := V_i(x_{i,k+1}) - \sum_{j \in \mathcal{E}_i \cup \{i\}} \gamma_{ij} V_j(x_{j,k}) - \psi \|d_{i,k}\|_2, \tag{58}$$

where $x_{i,k+1} = f_i\big(x_{i,k}, u_{i,k}, \{x_{j,k-s_{ij}}\}_{j \in \mathcal{E}_i}, d_{i,k}\big)$.

**Outside region.** Consider any continuous point

$$z_{i,k} \in (\mathcal{Z}_{i,k}^* \setminus \mathcal{S}_{R,i}) \times \mathcal{W}_i. \tag{59}$$

Let $\hat{z}_{i,k} \in \mathcal{D}_i^{\mathrm{out}}$ be a nearest grid point to $z_{i,k}$. Under uniform grid sampling with per-dimension step sizes $\Delta_{i,k}$, we have

$$\|z_{i,k} - \hat{z}_{i,k}\|_2 \le \tfrac{1}{2}\|\Delta_{i,k}\|_2 = \varepsilon_{i,k}^{\mathrm{out}}. \tag{60}$$

Let $L_{h_i}$ be a Lipschitz constant of $h_i(\cdot)$ on $(\mathcal{Z}_{i,k}^* \setminus \mathcal{S}_{R,i}) \times \mathcal{W}_i$ so that

$$|h_i(z_{i,k}) - h_i(\hat{z}_{i,k})| \le L_{h_i}\|z_{i,k} - \hat{z}_{i,k}\|_2 \le L_{h_i,\mathrm{upper}}\varepsilon_{i,k}^{\mathrm{out}}. \tag{61}$$

with

$$L_{h_i} \le L_{h_i,\mathrm{upper}} = p\, L_{V_i} + \max_{j \in \mathcal{E}_i \cup \{i\}} L_{V_j}, \tag{62}$$

where $L_{V_j}$ is a Lipschitz constant of $V_j$ on the corresponding projected domain. (Here the maximization over $s$ in (57) does not change the bound since each $V_j$ is Lipschitz with a constant $L_{V_j}$ independent of $s$.)

Similarly, let $L_{r_i}$ be a Lipschitz constant of $r_i(\cdot)$ on the same set so that

$$|r_i(z_{i,k}) - r_i(\hat{z}_{i,k})| \le L_{r_i}\|z_{i,k} - \hat{z}_{i,k}\|_2 \le L_{r_i}\varepsilon_{i,k}^{\mathrm{out}}. \tag{63}$$

with

$$L_{r_i} \le L_{r_i,\mathrm{upper}} = L_{V_i} L_{f_i} + \sum_{j \in \mathcal{E}_i \cup \{i\}} |\gamma_{ij}| L_{V_j} + \psi, \tag{64}$$

where we used that $d \mapsto \|d\|_2$ is 1-Lipschitz.

Now assume the Razumikhin logic is verified on $\mathcal{D}_i^{\mathrm{out}}$ with the robust margin:

$$h_i(\hat{z}_{i,k}) < -L_{h_i,\mathrm{upper}}\varepsilon_{i,k}^{\mathrm{out}} \ \lor \ r_i(\hat{z}_{i,k}) \le -\delta_{i,k}^{\mathrm{out}}, \qquad \forall \hat{z}_{i,k} \in \mathcal{D}_i^{\mathrm{out}}. \tag{65}$$

Equivalently,

$$h_i(\hat{z}_{i,k}) \ge -L_{h_i,\mathrm{upper}}\varepsilon_{i,k}^{\mathrm{out}} \ \Rightarrow \ r_i(\hat{z}_{i,k}) \le -\delta_{i,k}^{\mathrm{out}}. \tag{66}$$

Take any continuous $z_{i,k}$ such that the Razumikhin condition holds, i.e., $h_i(z_{i,k}) \ge 0$. By (61) and (60),

$$h_i(\hat{z}_{i,k}) \ge h_i(z_{i,k}) - L_{h_i,\mathrm{upper}}\varepsilon_{i,k}^{\mathrm{out}} \ge -L_{h_i,\mathrm{upper}}\varepsilon_{i,k}^{\mathrm{out}}. \tag{67}$$

Thus, (66) yields $r_i(\hat{z}_{i,k}) \le -\delta_{i,k}^{\mathrm{out}}$. Using (63), we obtain

$$r_i(z_{i,k}) \le r_i(\hat{z}_{i,k}) + L_{r_i}\varepsilon_i \le -\delta_{i,k}^{\mathrm{out}} + L_{r_i,\mathrm{upper}}\varepsilon_i^{\mathrm{out}}. \tag{68}$$

Under the validity condition

$$\delta_{i,k}^{\mathrm{out}} \ge L_{r_i,\mathrm{upper}}\varepsilon_{i,k}^{\mathrm{out}}, \tag{69}$$

we conclude $r_i(z_{i,k}) \le 0$, i.e.,

$$V_i(x_{i,k+1}) \le \sum_{j \in \mathcal{E}_i \cup \{i\}} \gamma_{ij} V_j(x_{j,k}) + \psi \|d_{i,k}\|_2, \tag{70}$$

for all $z_{i,k} \in (\mathcal{Z}_{i,k}^* \setminus \mathcal{S}_{R,i}) \times \mathcal{W}_i$ satisfying $h_i(z_{i,k}) \geq 0$.

**Inside region.** Now consider any continuous point $z_i \in \mathcal{S}_{R,i} \times \mathcal{W}_i$, and let $\hat{z}_i \in \mathcal{D}_i^{\text{in}}$ be a nearest grid point, so that $\|z_i - \hat{z}_i\|_2 \leq \varepsilon_i$. Assume the inside decrement inequality is verified on the sampled set:

$$r_i(\hat{z}_i) \leq -\delta_i^{\text{in}}, \qquad \forall \hat{z}_i \in \mathcal{D}_i^{\text{in}}. \tag{71}$$

Using (63) (with a Lipschitz constant $L_{r_i}$ valid on $\mathcal{S}_{R,i} \times \mathcal{W}_i$), we obtain

$$r_i(z_i) \leq r_i(\hat{z}_i) + L_{r_i,\text{upper}}\varepsilon_i^{\text{in}} \leq -\delta_i^{\text{in}} + L_{r_i,\text{upper}}\varepsilon_i^{\text{in}}. \tag{72}$$

where $\frac{1}{2}\|\Delta_i^{\text{in}}\|_2 = \varepsilon_i^{\text{in}}$.

Under the validity condition

$$\delta_i^{\text{in}} \geq L_{r_i,\text{upper}}\varepsilon_i^{\text{in}}, \tag{73}$$

we conclude $r_i(z_i) \leq 0$, i.e.,

$$V_i(x_{i,k+1}) \leq \sum_{j \in \mathcal{E}_i \cup \{i\}} \gamma_{ij} V_j(x_{j,k}) + \psi\|d_{i,k}\|_2, \qquad \forall z_i \in \mathcal{S}_{R,i} \times \mathcal{W}_i. \tag{74}$$

Combining the outside and inside parts shows that the verified sampled conditions lift to the corresponding continuous domains required by the two-stage verification condition in Theorem 5.3. This completes the proof. $\qquad\square$

### A.4. Proof of Theorem 5.10

*Proof.* Fix any node $j \in \widetilde{\mathcal{N}}$ and any admissible local delayed history of node $j$ in $\widetilde{\mathcal{I}}$, including $\tilde{x}_{j,k}$, its delayed neighbor states $\{\tilde{x}_{l,k-\tilde{s}_{jl}}\}_{l \in \widetilde{\mathcal{E}}_j}$, and the disturbance $\tilde{d}_{j,k}$.

Since $\widetilde{\mathcal{I}}$ is substructure-isomorphic to $\mathcal{I}$ in the sense of Definition 5.9, there exists a local map

$$\iota_j : \{j\} \cup \widetilde{\mathcal{E}}_j \to \mathcal{N} \tag{75}$$

such that

$$\tilde{f}_j = f_{\iota_j(j)}, \qquad \iota_j(\widetilde{\mathcal{E}}_j) = \mathcal{E}_{\iota_j(j)}, \qquad \tilde{s}_{jl} = s_{\iota_j(j)\iota_j(l)}, \quad \forall l \in \widetilde{\mathcal{E}}_j. \tag{76}$$

Define the corresponding local variables in $\mathcal{I}$ by the relabeling

$$x_{\iota_j(l),k-s} := \tilde{x}_{l,k-s}, \qquad \forall l \in \widetilde{\mathcal{E}}_j \cup \{j\}, \quad s \in \{0, \dots, \tau_{\max}\}, \tag{77}$$

and

$$d_{\iota_j(j),k} := \tilde{d}_{j,k}. \tag{78}$$

Then, using the preservation of local dynamics, neighbor sets, and delays, we obtain

$$\begin{aligned}
\tilde{x}_{j,k+1} &= \tilde{f}_j\Big(\tilde{x}_{j,k}, \{\tilde{x}_{l,k-\tilde{s}_{jl}}\}_{l \in \widetilde{\mathcal{E}}_j}, \tilde{d}_{j,k}\Big) \\
&= f_{\iota_j(j)}\Big(x_{\iota_j(j),k}, \{x_{\iota_j(l),k-s_{\iota_j(j)\iota_j(l)}}\}_{l \in \widetilde{\mathcal{E}}_j}, d_{\iota_j(j),k}\Big) \\
&= x_{\iota_j(j),k+1}.
\end{aligned} \tag{79}$$

Since $\mathcal{I}$ admits a vector Lyapunov–Razumikhin certificate $\{V_i\}_{i \in \mathcal{N}}$, the conditions in Theorem 4.1 hold for the node $\iota_j(j)$. Define the candidate certificate and gains for $\widetilde{\mathcal{I}}$ as

$$\widetilde{V}_j := V_{\iota_j(j)}, \qquad \widetilde{\gamma}_{jl} := \gamma_{\iota_j(j)\iota_j(l)}, \quad \forall l \in \widetilde{\mathcal{E}}_j \cup \{j\}. \tag{80}$$

By the consistency assumption in Theorem 5.10, for every $l \in \widetilde{\mathcal{E}}_j \cup \{j\}$, we have

$$\widetilde{V}_l = V_{\iota_l(l)} = V_{\iota_j(l)}. \tag{81}$$

We first verify the class-$\mathcal{K}_\infty$ bounds. Since $\widetilde{V}_j = V_{\iota_j(j)}$ and $x_{\iota_j(j),k} = \tilde{x}_{j,k}$, the bounds for $V_{\iota_j(j)}$ imply

$$\alpha_1(\|\tilde{x}_{j,k}\|_2) \le \widetilde{V}_j(\tilde{x}_{j,k}) \le \alpha_2(\|\tilde{x}_{j,k}\|_2). \tag{82}$$

Next, suppose that the Razumikhin condition for node $j$ in $\tilde{\mathcal{I}}$ holds, i.e.,

$$\max_{\substack{l \in \widetilde{\mathcal{E}}_j \cup \{j\} \\ s \in \{1,\ldots,\tau_{\max}\}}} \widetilde{V}_l(\tilde{x}_{l,k-s}) \le p\,\widetilde{V}_j(\tilde{x}_{j,k}). \tag{83}$$

Using (81) and the local relabeling, this condition is equivalent to

$$\max_{\substack{l \in \widetilde{\mathcal{E}}_j \cup \{j\} \\ s \in \{1,\ldots,\tau_{\max}\}}} V_{\iota_j(l)}(x_{\iota_j(l),k-s}) \le p\,V_{\iota_j(j)}(x_{\iota_j(j),k}). \tag{84}$$

Since $\iota_j(\widetilde{\mathcal{E}}_j) = \mathcal{E}_{\iota_j(j)}$, this is exactly the Razumikhin condition for node $\iota_j(j)$ in $\mathcal{I}$. Therefore, by the decrement condition of Theorem 4.1, we have

$$V_{\iota_j(j)}(x_{\iota_j(j),k+1}) \le \sum_{l \in \widetilde{\mathcal{E}}_j \cup \{j\}} \gamma_{\iota_j(j)\iota_j(l)} V_{\iota_j(l)}(x_{\iota_j(l),k}) + \psi \|d_{\iota_j(j),k}\|_2. \tag{85}$$

Using (79), (81), and the definitions of $\widetilde{V}_j$ and $\widetilde{\gamma}_{jl}$, the above inequality becomes

$$\widetilde{V}_j(\tilde{x}_{j,k+1}) \le \sum_{l \in \widetilde{\mathcal{E}}_j \cup \{j\}} \widetilde{\gamma}_{jl} \widetilde{V}_l(\tilde{x}_{l,k}) + \psi \|\tilde{d}_{j,k}\|_2. \tag{86}$$

Thus the Lyapunov–Razumikhin decrement condition holds for node $j$ in $\tilde{\mathcal{I}}$.

It remains to check the small-gain condition. Since the gains of $\tilde{\mathcal{I}}$ are copied from those of $\mathcal{I}$ through the local substructure correspondence, we have

$$\sum_{l \in \widetilde{\mathcal{E}}_j \cup \{j\}} \widetilde{\gamma}_{jl} = \sum_{l \in \widetilde{\mathcal{E}}_j \cup \{j\}} \gamma_{\iota_j(j)\iota_j(l)} = \sum_{r \in \mathcal{E}_{\iota_j(j)} \cup \{\iota_j(j)\}} \gamma_{\iota_j(j)r} \le 1 - \epsilon. \tag{87}$$

Because $j \in \widetilde{\mathcal{N}}$ was arbitrary, all nodes in $\tilde{\mathcal{I}}$ satisfy the class-$\mathcal{K}_\infty$ bounds, the Razumikhin decrement condition, and the small-gain condition. Therefore, $\tilde{\mathcal{I}}$ admits the vector Lyapunov–Razumikhin certificate $\{\widetilde{V}_j\}_{j \in \widetilde{\mathcal{N}}}$ with gains $\widetilde{\Gamma} = \{\widetilde{\gamma}_{jl}\}$. By Theorem 4.1, $\tilde{\mathcal{I}}$ is sISS. This completes the proof. $\square$

### A.5. Proof of Theorem 5.12

*Proof.* By Definition 5.11, there exists a permutation $\iota$ of $\mathcal{N}$ such that, for each $i \in \mathcal{N}$, (i) $f_i = f_{\iota(i)}$, (ii) $\iota(\mathcal{E}_i) = \mathcal{E}_{\iota(i)}$, and (iii) for each $j \in \mathcal{E}_i$, the delay satisfies $s_{ij} = s_{\iota(i)\iota(j)}$. By Theorem 5.3 in Gallian (Gallian, 2021), any permutation of a finite set has a finite order, i.e., there exists $M \in \mathbb{Z}_+$ such that $\iota^M(i) = i$ for all $i \in \mathcal{N}$.

Since $\mathcal{I}$ satisfies the delayed sISS conditions, it admits a vector Lyapunov-Razumikhin certificate $\{V_i\}_{i \in \mathcal{N}}$ with coupling gains $\Gamma = \{\gamma_{ij}\}$ (as in Theorem 4.1). For each $m \in \{0,\ldots,M-1\}$, consider the permuted family of functions

$$V_i^{(m)} := V_{\iota^m(i)}, \qquad i \in \mathcal{N}.$$

We claim that $\{V_i^{(m)}\}_{i \in \mathcal{N}}$ is also a valid delayed sISS certificate for $\mathcal{I}$. Indeed, fix any agent $i$ and any admissible delayed history $\{x_{i,k-s}\}_{s=0}^{\tau}$ together with neighbor histories $\{x_{j,k-s_{ij}}\}_{j \in \mathcal{E}_i}$ and disturbance $d_{i,k}$. Because $\iota$ preserves dynamics, neighbor sets, and link delays, we have the trajectory consistency

$$x_{\iota^m(i),k+1} = f_{\iota^m(i)}\Big(x_{\iota^m(i),k}, \{x_{\iota^m(j),k-s_{\iota^m(i)\iota^m(j)}}\}_{j \in \mathcal{E}_i}, d_{\iota^m(i),k}\Big)$$

$$= f_i\Big(x_{i,k}, \{x_{j,k-s_{ij}}\}_{j\in\mathcal{E}_i}, d_{i,k}\Big) = x_{i,k+1}, \tag{88}$$

where we used $f_{\iota^m(i)} = f_i$, $\iota^m(\mathcal{E}_i) = \mathcal{E}_{\iota^m(i)}$, and $s_{\iota^m(i)\iota^m(j)} = s_{ij}$ for all $j \in \mathcal{E}_i$. Consequently, applying the delayed sISS inequalities of Theorem 4.1 to the index $\iota^m(i)$ yields that, for some $\alpha_1, \alpha_2 \in \mathcal{K}_\infty$ and suitable gains,

$$\alpha_1(\|x_{i,k}\|_2) \ \leq \ V_{\iota^m(i)}(x_{i,k}) \ \leq \ \alpha_2(\|x_{i,k}\|_2), \tag{89}$$

and if the Razumikhin condition $\displaystyle\max_{\substack{j\in\mathcal{E}_i\cup\{i\} \\ s\in\{1,\dots,\tau_{\max}\}}} V_{\iota^m(j)}(x_{j,k-s}) \ \leq \ p\, V_{\iota^m(i)}(x_{i,k})$ holds, then

$$V_{\iota^m(i)}(x_{i,k+1}) \ \leq \ \sum_{j\in\mathcal{E}_i\cup\{i\}} \gamma_{\iota^m(i)\iota^m(j)}\, V_{\iota^m(j)}(x_{j,k}) + \psi\|d_{i,k}\|_2, \tag{90}$$

with the same delay indices $s_{ij}$ as in the original system due to (88). Thus each permuted family $\{V_i^{(m)}\}$ is a valid delayed sISS certificate.

Now define the max-aggregated certificate

$$\widetilde{V}_i(x) := \max_{m=0,\dots,M-1} V_{\iota^m(i)}(x), \qquad \forall i \in \mathcal{N}. \tag{91}$$

Since $\iota$ has order $M$, $\widetilde{V}_i$ is invariant under $\iota$, i.e., $\widetilde{V}_{\iota(i)} = \widetilde{V}_i$ for all $i \in \mathcal{N}$. In particular, any two structurally equivalent nodes $i$ and $j = \iota(i)$ share the same certificate function.

Pick $m^* \in \arg\max_m V_{\iota^m(i)}(x_{i,k+1})$. Then $\widetilde{V}_i(x_{i,k+1}) = V_{\iota^{m^*}(i)}(x_{i,k+1})$ and (89) implies the $\mathcal{K}_\infty$ bounds for $\widetilde{V}_i$. Moreover, using (90) and the fact that $V_{\iota^{m^*}(j)}(x_{j,k}) \leq \widetilde{V}_j(x_{j,k})$ for all $j$, we obtain

$$\begin{aligned}
\widetilde{V}_i(x_{i,k+1}) &= V_{\iota^{m^*}(i)}(x_{i,k+1}) \\
&\leq \sum_{j\in\mathcal{E}_i\cup\{i\}} \gamma_{\iota^{m^*}(i)\iota^{m^*}(j)}\, V_{\iota^{m^*}(j)}(x_{j,k}) + \psi\|d_{i,k}\|_2 \\
&\leq \sum_{j\in\mathcal{E}_i\cup\{i\}} \gamma_{\iota^{m^*}(i)\iota^{m^*}(j)}\, \widetilde{V}_j(x_{j,k}) + \psi\|d_{i,k}\|_2.
\end{aligned} \tag{92}$$

The corresponding permuted gain matrix $\widetilde{\Gamma} = \{\gamma_{\iota^{m^*}(i)\iota^{m^*}(j)}\}$ inherits the small-gain property from $\Gamma$. Therefore, $\{\widetilde{V}_i\}_{i\in\mathcal{N}}$ satisfies the delayed sISS conditions in Theorem 4.1, and by construction $\widetilde{V}_i = \widetilde{V}_j$ for any structurally equivalent nodes $i, j \in \mathcal{N}$. This completes the proof. $\square$

## B. CEGIS Loop

Algorithm 1 summarizes our CEGIS loop for neural vector Lyapunov-Razumikhin certificates synthesis. Starting from an initial dataset from nominal controller rollouts, we iteratively (i) train the candidate certificate $(\Gamma, V_{i\in\mathcal{N}})$ by minimizing the synthesis loss and (ii) verify the conditions in Theorem 5.3 over reachability-constrained delay domains. Whenever the verifier discovers violating delayed histories, they are appended to the training set to refine the certificate, and the loop terminates once no counterexamples are found.

## C. Illustrative Examples

Figure 4 shows three representative interconnection topologies: a star (a), a small tree (b), and a ring (c). An arrow $j \to i$ means $j \in \mathcal{E}_i$. For the star in (a), the hub is unique while all leaves are structurally equivalent (each interacts only with the hub), so verifying sISS for the hub and one leaf is sufficient and the remaining leaves follow by equivalence. Analogous reductions hold for the tree in (b) and the ring in (c).

## D. Experiment Settings

### D.1. System Dynamics for Application Environment

**Mixed-autonomy platoon.** We consider a mixed-autonomy platoon composed of CAVs $\Omega_\mathcal{C}$ and HDVs $\Omega_\mathcal{H}$, where $n := |\Omega_\mathcal{C} \cup \Omega_\mathcal{H}|$ denotes the total number of vehicles and $m := |\Omega_\mathcal{C}|$ denotes the number of CAVs. For each vehicle

---

**Algorithm 1** CEGIS Loop for Neural Vector Lyapunov–Razumikhin Certificates

---

1: **Input:** Initial dataset $\{\mathcal{D}_{i,k}\}_{i\in\mathcal{N},k\in[0,\cdots,T_R]}$, $\{\mathcal{D}_i^{\text{out}}\}_{i\in\mathcal{N}}$ (e.g., nominal-controller rollouts), initial coupling gains $\Gamma = \{\gamma_{ij}\}$, Lyapunov-Razumikhin functions $\{V_i\}_{i\in\mathcal{N}}$.
2: **Output:** Verified certificate $(\Gamma, \{V_i\}_{i\in\mathcal{N}})$.
3: **Initialize:** Initialize $\Gamma$ and $\{V_i\}_{i\in\mathcal{N}}$.
4: **repeat**
5:     Train $\Gamma$ and $\{V_i\}_{i\in\mathcal{N}}$ by minimizing the synthesis loss (i.e., Eq. (9)).
6:     Verify the delayed Razumikhin conditions in Theorem 5.8 over the reachability-constrained delay domains.
7:     **if** counterexamples violating the conditions are found **then**
8:         Augment $\{\mathcal{D}_{i,k}\}_{i\in\mathcal{N},k\in[0,\cdots,T_R]}$, $\{\mathcal{D}_i^{\text{out}}\}_{i\in\mathcal{N}}$ with the counterexamples.
9:     **end if**
10: **until** no counterexamples are found after verification.

---

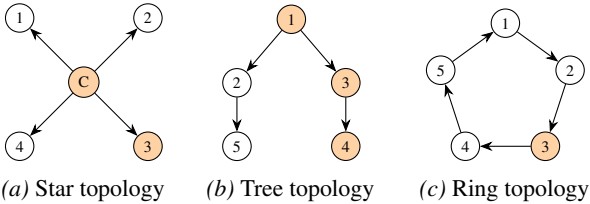

*(a)* Star topology      *(b)* Tree topology      *(c)* Ring topology

*Figure 4.* Representative network topologies demonstrating structural equivalence among nodes. Highlighted nodes are the selected representatives to verify.

$i \in \Omega_{\mathcal{C}} \cup \Omega_{\mathcal{H}}$, under a sampling period $T > 0$, the discrete-time longitudinal dynamics are

$$s_{i,k+1} = s_{i,k} + T\big(v_{i-1,k} - v_{i,k}\big), \tag{93}$$

$$v_{i,k+1} = v_{i,k} + T \begin{cases} u_{i,k}, & i \in \Omega_{\mathcal{C}}, \\ \mathbb{F}_i(s_{i,k}, v_{i,k}, v_{i-1,k}), & i \in \Omega_{\mathcal{H}}, \end{cases} \tag{94}$$

where $s_{i,k}$ and $v_{i,k}$ denote the spacing and velocity at step $k$. The CAV input $u_{i,k}$ is generated by an RL policy (Zhou et al., 2026), whereas each HDV follows an unknown car-following model.

**Drone formation.** We consider a leader-follower formation in $\mathbb{R}^3$ consisting of one leader and $N_f$ followers. Let $p_\ell, v_\ell, u_\ell \in \mathbb{R}^3$ denote the leader's position, velocity, and acceleration, and let $p_i, v_i, u_i$ denote the corresponding quantities for follower $i$. The leader tracks a predefined trajectory:

$$p_{\ell,k+1} = p_{\ell,k} + T\, v_{\ell,k}, \tag{95}$$

$$v_{\ell,k+1} = v_{\ell,k} + T\, u_{\ell,k}. \tag{96}$$

Each follower evolves according to

$$x_{i,k+1} = \mathrm{F}_i\big(x_{i,k}, u_{i,k}, x_{i,k}^r\big), \tag{97}$$

where $x_{i,k} = [p_{i,k}, v_{i,k}]^\top$ and $x_{i,k}^r$ denotes the reference state (from the leader or the preceding drone). The true dynamics $\mathrm{F}_i$ are unknown and are estimated from data. The follower control input is generated by a neural policy trained via supervised learning to imitate a known controller.

**Microgrid.** We consider a microgrid with $n$ voltage-source inverters indexed by $\mathcal{N} = \{1, \ldots, n\}$, interconnected in a radial topology with neighbor sets $\mathcal{E}_i$. Each inverter $i$ has phase angle $\delta_{i,k}$, voltage magnitude $U_{i,k} > 0$, and frequency $\omega_{i,k}$, with state $x_{i,k} = (\delta_{i,k}, \omega_{i,k}, \xi_{i,k})$. The active power injection at node $i$ is

$$P_{i,k} = P_{L,i} + \sum_{j\in\mathcal{E}_i} \alpha_{ij} \sin(\delta_{i,k} - \delta_{j,k}), \qquad \alpha_{ij} = |B_{ij}|U_{i,k}U_{j,k}, \tag{98}$$

where $P_{L,i}$ is the load demand. The closed-loop dynamics satisfy

$$\delta_{i,k+1} = \delta_{i,k} + T\omega_{i,k}, \tag{99}$$

$$\omega_{i,k+1} = \omega_{i,k} + \frac{T}{\tau_i}\big[-(\omega_{i,k}-\omega^*) - \eta_i(P_{i,k}-P_i^*) + \xi_{i,k}\big], \tag{100}$$

$$\xi_{i,k+1} = \xi_{i,k} + Tu_{i,k}, \tag{101}$$

where $\tau_i > 0$ is the frequency-loop time constant, $\eta_i > 0$ is the active-power droop gain, $P_i^*$ is the active-power setpoint, and $\omega^*$ denotes the nominal frequency. The secondary control input is updated by a neural policy:

$$u_{i,k} = \pi_{\theta_i}\big(x_{i,k}, \{x_{j,k}\}_{j\in\mathcal{E}_i}\big). \tag{102}$$

### D.2. Implementation Details

**Hardware and System Configuration.** All experiments were conducted on a Linux workstation running Ubuntu 24.04, equipped with an Intel Core Ultra 9 285K CPU (24 cores), 125 GB RAM, and two NVIDIA GeForce RTX 5090 GPUs (32 GB memory each). The system uses Linux kernel 6.14.0 with CUDA 12.8 and NVIDIA driver 570.153.02.

**Training Details.** Each CEGIS loop is trained for at most 100 epochs, for up to 100 iterations. The learning rate is initialized at 0.001 and decayed according to a predefined schedule. The initial dataset consists of 30000 trajectories of length 50, split into 80% for training and 20% for validation, with a batch size of 32. Both the Lyapunov function and the controller use three-layer fully connected networks with 64 hidden units per layer. Each runtime experiment is run three times, and we report the mean $\pm$ standard deviation. For simplicity, we set the disturbance to $d_{i,k} \equiv 0$, so the reachable sets (and the corresponding rollouts used to construct them) are generated by simulating the closed-loop nominal dynamics.

**Hyperparameter.** In the loss function (Eq. (9)), we set $\epsilon_p = 10^{-3}$ and $\epsilon_d = 10^{-6}$, with weighting coefficients $w_{\text{imi}} = 1$, $w_p = 4000$, and $w_d = 2000$. For the verification in Eq. (22), we use $p = c = 1.01$, $\rho = 0.95$, and $R = 0.15$ across all three environments. We further set $\Delta_{i,k} = 0.05$, $\Delta_i^{\text{in}} = 0.01$, and $T_R = 50$.

## E. Future Work

Several directions are of immediate interest. First, the current analysis can be extended to time-varying communication delays with a time-varying upper bound, allowing the Razumikhin conditions and the reachability-constrained domains to adapt online to $\tau_{\max}(k)$. Second, safety guarantees in delayed environments can be incorporated by jointly synthesizing and verifying safety certificates alongside the delayed-sISS certificate, enabling certified safe operation under bounded delays and disturbances. Third, tighter reachability over-approximations can be developed to reduce conservatism in the verification domains, for example by exploiting monotonicity/contractivity properties and compositional set propagation, thereby improving scalability and reducing false counterexamples in the CEGIS loop.

