# OpenReview forum: "Neural Vector Lyapunov–Razumikhin Certificates for Delayed Interconnected Systems"
_ICML.cc/2026/Conference — ICML 2026 regular_

### Official Review · Reviewer_pAxK · 2026-02-20

**Soundness:** 3
**Presentation:** 3
**Significance:** 3
**Originality:** 3
**Overall Recommendation:** 4
**Confidence:** 1

**Summary:**

The paper proposes a novel framework to synthesize and verify neural vector Lyapunov-Razumikhin certificates for discrete-time delayed interconnected systems, aiming to provide formal scalable input-to-state stability guarantees for large-scale networks with communication delays.

**Compliance With Llm Reviewing Policy:**

Affirmed.

**Final Justification:**

The paper presents a well-executed framework with clear exposition and empirical validation, and the rebuttal adequately clarifies concerns on assumptions and robustness. Overall, I am not an expert in this field and I maintain a relatively positive recommendation with low confidence.

**Key Questions For Authors:**

1. Can the auther gives more explanation about Assumption (ii) in Theorem 4.1? I wonder why this assumption is reasonable.

2. In the experimental evaluation, how does the learned neural certificate perform when the actual system dynamics deviate significantly from the nominal policy used during the imitation learning phase?

**Limitations:**

yes

**Strengths And Weaknesses:**

Strengths

1: The paper is well-written and easy to follow. Even for me who is not experts in this specific field, the problem formulation is presented clearly and logically.

2: The authors provide a comprehensive set of detailed experiments that effectively verify the paper's conclusions.

Weaknesses

1: The two-stage verification strategy and the reliance on reachability over-approximations may introduce conservatism, potentially leading to false counterexamples or restricted stability regions.

2: Assumption (ii) in Theorem 4.1 seems to be strict. It is unclear whether this condition can be realistically satisfied in practical application setups.

---

> ### Author Rebuttal · Authors · 2026-03-31
>
> We sincerely thank the reviewer for the insightful and constructive comments, and for recognizing the theoretical contribution, the quality of the paper writing, and the solid experimental results of our proposed framework. Our point-by-point response is as follows:
>
> **1.Explanation on conservatism (W1).** The proposed two-stage verification may introduce some conservatism in practice, but this is a necessary trade-off for enabling verification over an infinite time horizon. In particular, Stage 1 alone only characterizes finite-step reachable sets and therefore cannot by itself guarantee the desired property for all future time, which makes Stage 2 necessary. At the same time, this conservatism is controllable. Specifically, it is governed by the reachability radius $R$ in Eq. (18). If one wishes to reduce conservatism, $R$ can be adjusted so that Stage 1 provides a tighter reachable-set approximation and hence covers a larger portion of the verification task before passing to Stage 2. When $R$ is chosen sufficiently small, the induced relaxation can be made correspondingly small. Moreover, the reachability step is introduced to restrict verification to reachable subsets rather than the full delayed domain, which typically reduces the verification region in practice. Therefore, the two-stage design offers a tunable trade-off between conservatism and verification coverage. We will clarify this trade-off more explicitly in the revised manuscript.
>
> **2.Clarification of decremental condition in Theorem 4.1 (W2, Q1).** We would like to clarify that Condition (ii) is not imposed as a property that must be satisfied a priori by an arbitrary delayed interconnected system. In our framework, it is enforced by construction through the joint design of the controller, Lyapunov functions, and coupling gains, and is then certified through verification. Specifically:
>
> First, Razumikhin-type conditions are a standard tool in delay-system analysis [1], especially under bounded-delay assumptions, which are common in practice. In our setting, the key requirement is that the influence of delayed neighboring states remain dominated by the local Lyapunov decrease. When the delay is bounded and the delay-free closed-loop dynamics is sufficiently dissipative, the Razumikhin-type condition can be satisfied.
>
> Second, in our framework, the Razumikhin condition is not treated as a passive property that the original system must satisfy a assumption. Rather, it is enforced through the closed-loop design: the controller, Lyapunov function, and coupling gains are learned jointly so that the delayed interconnection effect is dominated by the desired Lyapunov decrease. In this sense, the condition is made to hold by learning and verification, rather than assumed to arise automatically.
>
> Third, we clarify that the small-gain condition for $\Gamma$ in our paper is not a strict requirement. Small-gain condtion is a standard and widely used tool for studying stability of large-scale interconnected systems. Moreover, the small-gain condition in our paper does not imply that the underlying physical system needs to be weakly coupled [2]. In our formulation, the small-gain condition is imposed at the Lyapunov level, through the decremental relation among subsystems (see Eq. (7)), rather than at the level of the physical dynamics themselves. Accordingly, the entries of the gain matrix quantify how Lyapunov functions interact in the decrease inequality, rather than the actual coupling strength in the system equations. Hence, satisfying the small-gain condition does not mean that the physical interconnection is weak or that the system is nearly decoupled. Moreover, in our framework, this condition is enforced by construction through the parameterization of the coupling matrix $\Gamma$ (see the discussion immediately above Eq. (9)).
>
> We will clarify this points more explicitly in the revised manuscript.
>
> [1] Schlaginhaufen et al. (2021). Learning stable deep dynamics models for partially observed or delayed dynamical systems. NeurIPS.
>
> [2] Cui et al. (2021). Asymptotic trajectory tracking of autonomous bicycles via backstepping and optimal control. IEEE Control Systems Letters, 6, 1292-1297.
>
> **3.Explanation on robustness to dynamics mismatch (Q2).** We thank the reviewer for this insightful question. If the deviation between the actual dynamics and the nominal policy used during imitation learning can be upper-bounded, then it can be considered into the verification margin Eq. (22), so the certificate remains valid in a correspondingly robust sense. In this case, the main effect is increased conservatism rather than loss of soundness. If the deviation is too large to be covered by the margin, then the original certificate may no longer apply directly, and re-verification and re-training would be needed. We will include this point in the conclusion and future work section of the revised manuscript.

---

> > ### Author Rebuttal · Reviewer_pAxK · 2026-04-01
> >
> > My concerns have been adequately addressed. Thank you. I have also read the authors’ responses to other reviewers and decide to maintain a positive score with low confidence.

---

> > > ### Author Response · Authors · 2026-04-01
> > >
> > > Thank you for your time and consideration. We sincerely appreciate your positive assessment.

---

### Official Review · Reviewer_FsgE · 2026-02-23

**Soundness:** 3
**Presentation:** 2
**Significance:** 3
**Originality:** 3
**Overall Recommendation:** 2
**Confidence:** 4

**Summary:**

Authors propose a new method for synthesis and verification of neural vector Lyapunov-Razumikhin certificates for delayed ISs. They take inspiration from previous work, and utilize ReLU function to encode conditions, then they develop a verification method based on finite horizon reachability and sampling. Finally, authors demonstrate their method on multiple systems.

**Compliance With Llm Reviewing Policy:**

Affirmed.

**Key Questions For Authors:**

see weaknesses.

**Limitations:**

Partial, see weaknesses.

**Strengths And Weaknesses:**

Strengths:

1- Paper is rigorous and easy to follow, concepts are laid out in an intuitive manner.

2- Authors do a good job of providing formal-ish guarantees for a difficult problem.

3- Theoretical results are interesting.

4- The experimental section is strong in demonstrating the practical viability of the verification framework. In particular, the runtime comparisons clearly show that combining reachability constrained domains with structural reuse is essential for scalability, and the results convincingly illustrate near invariance of verification cost with respect to network size. This is an important and well supported empirical message, and it aligns closely with the core technical contributions of the paper.

At the same time, it may be more effective for the authors to focus the experimental narrative primarily on verification and scalability rather than controller performance. The method is fundamentally a certification and verification framework; it does not explicitly optimize tracking error or integrate performance objectives beyond regularization toward a nominal policy. As a result, the observed improvements in RMSE appear to be secondary effects rather than direct outcomes of the proposed theory. Emphasizing reachability reduction, scalability, and formal guarantees would therefore present a clearer and more coherent experimental story.


Weaknesses:

1- In order to make their approach viable, authors resort to sampling based techniques. This is an understandable trade-off, however I wished authors would frame it accordingly. Formal guarantee is different than approximate ones, this should have been stated more clearly in the paper.

2-While the paper states that there are no prior neural certificates for delayed large scale systems, the novelty may still appear incremental to some readers. At a high level, the work can be seen as extending vector Lyapunov ISS ideas to delayed systems using Razumikhin conditions and adapting existing neural Lyapunov learning and sampling based verification pipelines to delay embedded domains.

Most of the core ingredients, including vector ISS theory, Razumikhin methods, reachability reduction, CEGIS style refinement, and Lipschitz margin arguments, are well known individually. The main contribution lies in integrating these components into a unified framework and preserving scalability in the presence of delays.

To strengthen the contribution, the paper would benefit from a clearer discussion of its mathematical novelty. It would help to clarify what is fundamentally new beyond adapting known tools, whether Theorem 4.1 is structurally new or a reformulation of classical Razumikhin arguments in vector form, and how the results compare formally to Lyapunov Krasovskii based scalable approaches. Without this positioning, the work may be viewed as a strong engineering synthesis rather than a clear conceptual advance.

3-The sampling based guarantee relies on several strong assumptions. It requires known Lipschitz constants for the system dynamics, neural Lipschitz bounds that can be loose in practice, and uniform grid sampling over the delay domain, which can grow exponentially with the delay horizon and neighborhood size. Although the proposed structural reduction helps mitigate this blow up in practice, the theoretical worst case complexity remains severe.

The paper would be strengthened by a clearer discussion of computational scaling, including how complexity depends on the maximum delay and neighborhood size. It would also be useful to comment on how tight the Lipschitz bounds are in practice and how this affects verification margins. Finally, comparing against MILP based neural verification baselines would help contextualize the efficiency and conservativeness of the proposed approach.

---

> ### Author Rebuttal · Authors · 2026-03-31
>
> We sincerely thank the reviewer for the constructive and professional comments, and for recognizing the theoretical contribution, the quality of the paper writing, and the solid experimental results. Our point-by-point response is as follows:
>
> **1. Clarification on verification (W1).** We clarify that our guarantee is formal rather than approximate. While verification is performed on sampled states, the combination of Lipschitz continuity and sampling resolution allows the results to be rigorously extended to the entire domain. In the revised manuscript, we will clarify that our approach is a sampling-based formal verification method, consistent with existing literature that uses discretization with Lipschitz assumptions [1].
>
> [1] Anand et al. (2023). Formally verified neural network control barrier certificates for unknown systems. IFAC-PapersOnLine.
>
> **2. Clarification on contributions (W2).** We clarify that our work provides non-trivial theoretical contributions beyond a combination of existing techniques.
>
> First, Theorem 4.1 is not a direct extension of classical results. While standard Razumikhin results address single-system stability, we establish scalable Input-to-State Stability for interconnected systems. This requires capturing disturbance propagation such that stability bounds remain independent of network size, which is a property absent in single-system settings. We chose the Razumikhin form specifically for its compatibility with neural certificate parameterization, as Lyapunov-Krasovskii functionals are significantly harder to verify formally.
>
> Second, Theorems 5.3 and 5.8 provide the first formal verification framework for delayed neural certificates via a two-stage reachability-based procedure and to the best of our knowledge, are the first to enable verification over an infinite time horizon in this setting. Furthermore, Theorems 5.10 and 5.12 provide the scalability analysis for delayed large-scale systems. These results contribute distinct mathematical contributions to formal guarantees, rather than a mere engineering synthesis of existing tools.
>
> We will revise the manuscript to emphasize these contributions more clearly.
>
> **3.Clarification on Lipschitz bound (W3 part 1).** We clarify that our framework does not require the explicit system dynamics when the Lipschitz constant is estimated from data via set-membership identification [2]. We also note that Lipschitz continuity assumption for system dynamics is commonly adopted for verifying neural certificates (e.g., [1]).
>
> Moreover, we added a five-vehicle platoon comparison of NN Lipschitz bounding methods:
> |Metric|ProductNorm|LipSDP[3]|ECLipsE[4]|
> |---|---:|---:|---:|
> |Lipschitz bound for agent 1|1.59|0.52|0.46|
> |Verification|Time Out|Pass|Pass|
>
> LipSDP and ECLipsE ensure successful certification and norm-product bounds fail due to conservatism. The detailed experiments will be added to the manuscript.
>
> [2] Calliess et al. (2017). Lipschitz optimisation for Lipschitz interpolation. In 2017 American Control Conference.
>
> [3] Fazlyab et al.. (2019). Efficient and accurate estimation of Lipschitz constants for deep neural networks. NeurIPS.
>
> [4] Xu et al. (2024). Eclipse: Efficient compositional lipschitz constant estimation for deep neural networks. NeurIPS.
>
> **4.Discussion on MILP-based method (W3 part 2).** We did consider MILP-based neural network verification tools such as Marabou with Gurobi solver. However, applying such methods in our setting would require first approximating the physical system dynamics by a neural network surrogate. This introduces an additional mismatch between the surrogate model and the true system dynamics, and obtaining guarantees for the original physical system would then require quantifying this approximation error and incorporating additional conservative margins into the verification procedure. To avoid this extra modeling layer, we instead adopt a sampling-based verification framework combined with Lipschitz-based bounds. We will clarify this point in the revised paper.
>
> **5.Discussion of computational scaling (W3 part 3).** We will explicitly state the scaling in the revision. Without reachability-based reduction, if each state is gridded with $M$ samples, then the total number of verification points over all $N$ subsystems scales as $\sum_{i=1}^N M^{(\tau_{\max}+1)(n_i+\sum_{j\in\mathcal E_i} n_j)}$, which grows rapidly with the delay, neighborhood size, and number of agents. Our reachability-based reduction substantially decreases the number of points that need to be verified by restricting verification to reachable subsets rather than the full delayed domain. In addition, Theorems 5.10 and 5.12 further improve scalability with respect to the number of agents through structural reuse [5]. To address local state dimensionality, we will explore monotonicity-based reduction in future work.
>
> [5] Alavi et al. (2025). Neural barrier certificates for monotone systems. IEEE Control Systems Letters.

---

> > ### Author Rebuttal · Reviewer_FsgE · 2026-03-31
> >
> > See my comment.

---

> > > ### Author Response · Authors · 2026-04-01
> > >
> > > After reading our rebuttal that addressed all comments (see below), the reviewer **reduced the score substantially from 4 to 2 without providing any justification beyond “see my comment”**. However, **the comments and subscores remain unchanged from the original review, which has been overall positive**, including comments such as "paper is rigorous", "good job of providing formal-ish guarantees for a difficult problem", "interesting" theoretical results, and experiments "strong in demonstrating the practical viability of the verification framework".
> > >
> > > In light of this, we believe such a substantial reduction in score would require the identification of a new and serious technical flaw that was not apparent in the original review, which is however not the case. It is therefore difficult for us to understand **how essentially the same set of comments could support a score of 4 before rebuttal, but only a 2 afterward, especially given that the reviewer maintained a high confidence level (4) throughout.** Without newly identified issues, such a downgrade appears unreasonable and raises concerns regarding the consistency and responsibility of the evaluation.
> > >
> > > We would therefore sincerely appreciate it if the reviewer could **clarify whether a new major technical flaw was identified**, and if so, what it is. We are not challenging the reviewer’s right to disagree with us on the merits. Rather, we aim to better understand the basis for the revised score, so that we can provide a focused and meaningful response and ensure that the evaluation process is fair and transparent.
> > >
> > > ---
> > >
> > > **Summary of previous rebuttal**. We summarize how each comment has been addressed, which we believe has resolved the reviewer's concerns and should not lead to a substantial reduction in score.
> > >
> > > **1. (W1).** We have **clarified that our guarantee is formal rather than approximate** with Theorem 5.8. The verification condition is first checked on a finite set of samples with sound robustness margins, and is then extended to the entire continuous domain under Lipschitz assumptions. Consequently, the guarantee holds over the full domain rather than relying on the sampled points.
> > >
> > > **2. (W2).** We have **clarified our mathematical novelty**, which is neither incremental nor merely an engineering synthesis. To the best of our knowledge, **we are the first to**
> > >
> > > **(i) introduce a sISS condition for large-scale interconnected systems with delays (Theorem 4.1)**. Theorem 4.1 is not a restatement of a classical Razumikhin theorem in vector form. Its novelty lies in reducing verification to localized subsystem-level inequalities by establishing a delay-dependent sISS condition together with a corresponding small-gain condition. Unlike traditional ISS theorems that suffer from the curse of dimensionality in verification, our sISS framework is specifically designed to scale with the number of agents. **Compared to Lyapunov-Krasovskii functionals, which are difficult to parameterize with neural networks due to their infinite-dimensional functional form**, our approach uses a more computationally tractable representation for formal verification;
> > >
> > > (ii) derive a **novel reachability-based sufficient condition** that reduces the formal verification domain for delayed interconnected systems while preserving soundness, and apply it to certify the sISS condition with neural certificates; and
> > >
> > > (iii) develop a **novel structure-reuse-based scalability condition** that enables sound certificate reuse across subsystems with identical local structures.
> > >
> > > **3. (W3 part 1).** We have clarified that (i) **the Lipschitz constant of the system dynamics does not need to be known** a priori, as it can be upper-bounded via set-membership identification; and (ii) **the Lipschitz constants of the neural networks are tractable** for verification using state-of-the-art bounding methods such as LipSDP and ECLipsE. We also added **empirical results** showing that tighter estimators improve verification feasibility.
> > >
> > > **4. (W3 part 2).** We have clarified that existing MILP-based neural verification methods typically assume a neural-network model of the closed-loop dynamics, whereas our setting avoids this additional model learning step and its associated approximation error. Hence, the two settings are not directly comparable.
> > >
> > > **5. (W3 part 3).** We provided a **discussion of computational scalng** with respect to the maximum delay and the number of agents. If each state dimension is gridded with $M$ samples, the total number of verification points scales as $\sum_{i=1}^N M^{(\tau_{\max}+1)(n_i+\sum_{j\in\mathcal E_i} n_j)}$. In our framework, reachability reduction mainly decreases the effective grid size $M$, while structure reuse reduces the number of subsystems $N$ that need to be verified.

---

### Official Review · Reviewer_1s4A · 2026-03-11

**Soundness:** 2
**Presentation:** 2
**Significance:** 3
**Originality:** 2
**Overall Recommendation:** 5
**Confidence:** 3

**Summary:**

The paper proposes a scalable stability certification for delayed interconnected systems using neural Lyapunov-Razumikhin certificates.  The authors study discrete-time interconnected systems with communication delays and proposes a synthesis and verification framework for neural vector Lyapunov-Razumikhin certificates. The authors combine local Lyapunov certificate learning, reachability-constrained verification, and leverage nodal structure of the agents to improve tractability on large-scale systems. Several numerical results are demonstrated on platoons, drone formations, and microgrids.

**Compliance With Llm Reviewing Policy:**

Affirmed.

**Final Justification:**

The authors have replied reasonably to my comments and questions. I do not have any further questions and concerns.

**Key Questions For Authors:**

[Q1] the literature review should be expanded and the novelty more carefully positioned relative to prior work on stable-by-design neural control and unconstrained parameterizations. A comparison with these approaches and how they cannot be employed in the proposed method:
* Furieri, L., Galimberti, C. L., Zakwan, M., and Ferrari-Trecate, G. (2022). Distributed neural network control with dependability guarantees: a compositional port-Hamiltonian approach. L4DC.
* Zakwan, M. and Ferrari-Trecate, G. (2024). Neural port-Hamiltonian models for nonlinear distributed control: An unconstrained parametrization approach. arXiv:2411.10096.
* Manchester, I. R., Wang, R., and Barbara, N. H. (2026). Neural Networks in the Loop: Learning with Stability and Robustness Guarantees. Annual Review of Control, Robotics, and Autonomous Systems.
* Barbara, N. H., Wang, R., Megretski, A., and Manchester, I. R. (2025). React to surprises: Stable-by-design neural feedback control and the Youla-REN. arXiv:2506.01226.
*
[Q2] What is the main rationale for choosing Lyapunov-Razumikhin functions instead of Lyapunov-Krasovskii functionals? Much of the delay-systems literature in control uses Lyapunov-Krasovskii constructions. A more explicit discussion of this design choice, including its benefits and limitations in the present setting, would strengthen the paper.

[Q3] Eqs. (16) and (17) would be easier to understand with a graphical illustration of the reachability-constrained delay domains. A figure analogous to Figure 1 would be helpful. A clarification is important.

[Q4] The discussion of Lipschitz estimation is incomplete. There are several relevant methods for bounding neural-network Lipschitz constants that should be cited, for example:
* Fazlyab, M., Robey, A., Hassani, H., Morari, M., and Pappas, G. (2019). Efficient and accurate estimation of Lipschitz constants for deep neural networks. NeurIPS.
Can the authors comment on the conservatism of the Lipschitz constant using different methods and how they affect their verification framework.

**Limitations:**

Yes

**Strengths And Weaknesses:**

Strengths:
* The problem considered is important. Stability certification for large-scale interconnected systems with delays is a relevant and challenging topic, and the paper addresses it with a well-structured framework.
* The attempt to exploit nodal structure and certificate reuse to improve scalability is interesting and meaningful. The paper is also clearly organized at a high level.


Weaknesses:
* The literature review is incomplete. While the authors emphasize the lack of direct results for delayed interconnected systems, they do not sufficiently position their work relative to a stream of stability-by-design and unconstrained-parameterization approaches for nonlinear neural control. In particular, several relevant works on neural control with structural stability guarantees are missing, and these are important for properly assessing novelty.

* The notation for delayed systems needs improvement. In Eq. (1), the delayed system is introduced without a proper discussion of the initial conditions. For time-delay systems, the state is typically infinite-dimensional, and initial conditions should be defined as a functional over the delay interval.

* The method is not fully model-free. In Remark 5.7, the authors state that the Lipschitz constant of the dynamics is derived a priori from the analytic model by bounding the Jacobian norm using gridding based approaches. This means that the verification pipeline relies on model knowledge. That dependence should be stated more explicitly, especially since the paper motivates the framework in part by the difficulty of obtaining accurate models.

* The scalability claims should be qualified more carefully. Although the method leverages network structure and certificate sharing, the verification still relies heavily on gridding. As acknowledged in Section 5.2 and Theorem 5.8, the sampling complexity grows poorly with the state dimension. Thus, while the approach may scale with the number of nodes under favorable symmetry/isomorphism assumptions, it does not appear to scale well with local state dimension. Moreover, the assumption that relevant substructures are isomorphic is strong, and the paper would benefit from a broader discussion of practically important topologies where this assumption is realistic.

* The empirical performance gains are mixed. In particular, for the drone formation benchmark, the improvement in RMSE over the baselines is numerically negligible, as the paper itself effectively shows in Section 6.2. Therefore, the experimental results support the verification-efficiency claim more strongly than the control-performance claim.

---

> ### Author Rebuttal · Authors · 2026-03-31
>
> We sincerely thank the reviewer for the constructive and professional comments, and for recognizing the theoretical contributions and the quality of the paper writing. Our point-by-point response is as follows:
>
> **1.Discussion of related work (W1,Q1).** We agree that positioning our work within the broader context of stability-by-design is important for a complete literature review. Following the reviewer’s suggestion, we have revised the Introduction and Related Work sections accordingly. In particular, we clarify the relationship between our method and the following two relevant lines of work.
>
> First, Furieri et al. (2022) and Zakwan et al. (2024) use port-Hamiltonian structures for dissipativity-based stability. However, these rely on structural assumptions not met in our cases and do not account for communication delays.
>
> Second, Manchester et al. (2026) and Barbara et al. (2025) provide stability-by-design using Youla-type Parameterization for single nonlinear systems, but do not address interconnected systems or delay dynamics.
>
> **2.Notation of initial conditions for delayed system (W2).** We thank the reviewer for pointing out that the initial conditions of the delayed system should be stated more precisely. In the revised manuscript, we clarify this point as follows.
>
> Define $\tau_{\max}:=\max_{i,j}\tau_{ij}$. To properly specify the delayed dynamics, the initial condition is given by a history sequence
> $$
> x_{i,k}=\phi_i(k), \qquad k\in\lbrace-\tau_{\max},-\tau_{\max}+1,\dots,0\rbrace,
> $$
> for each $i\in\{1,\dots,N\}$,with $\phi_i:\lbrace-\tau_{\max},-\tau_{\max}+1,\dots,0\rbrace\to\mathbb R^{n_i}$ denotes the initial history of subsystem $i$.
>
> Since our setting is discrete-time systems, this history sequence is similar of the initial functional used in continuous-time delay systems.
>
> **3.Clarification on Lipschitz assumption (W3,Q4).** We clarify that our framework does not require a mathematical system dynamics model when the Lipschitz constant is estimated from data via set-membership identification [1]. We also note that Lipschitz continuity assumption for system dynamics is commonly adopted in learning-based control for deriving stability certificates (e.g., [2]).
>
> Moreover, we added a five-vehicle platoon comparison of NN Lipschitz bounding methods:
> |Metric|ProductNorm|LipSDP[3]|ECLipsE[4]|
> |---|---:|---:|---:|
> |Lipschitz bound for agent 1|1.59|0.52|0.46|
> |Verification|Time Out|Pass|Pass|
>
> Tighter methods (LipSDP, ECLipsE) ensure successful certification where norm-product bounds fail due to conservatism.
>
> The explanation and detailed experiments will be added to the revised manuscript.
>
> [1] Calliess et al. (2017). Lipschitz optimisation for Lipschitz interpolation. In 2017 American Control Conference.
>
> [2] Anand et al. (2023). Formally verified neural network control barrier certificates for unknown systems. IFAC-PapersOnLine.
>
> [3] Fazlyab et al.. (2019). Efficient and accurate estimation of Lipschitz constants for deep neural networks. NeurIPS.
>
> [4] Xu et al. (2024). Eclipse: Efficient compositional lipschitz constant estimation for deep neural networks. NeurIPS.
>
> **4.Clarification on scalability (W4).**  First, we clarify that while our framework enables delayed-system verification, it currently suits modest local state dimensions. Future work includes leveraging monotonicity [5] to reduce verification to interval endpoints.
>
> Second, the isomorphism assumption remains relevant for many large-scale systems with repeated local structures. As discussed in Appendix C, this applies not only to ring and tree topologies, but also to more complex networks formed by combining such recurring structures, such as large-scale microgrids and robotic formations. We will clarify this point and expand Appendix C accordingly.
>
> [5] Alavi et al. (2025). Neural barrier certificates for monotone systems. IEEE Control Systems Letters.
>
> **5.Clarification on evaluation performance (W5).** The relatively small RMSE gain in the drone formation benchmark is mainly due to its simple topology and weak coupling, since each UAV is only influenced by its predecessor. As a result, the room for improvement is limited. By contrast, our method yields much larger gains in the more strongly coupled platoon and microgrid benchmarks. The corresponding dynamics are provided in Appendix B, and we will clarify this point in the experimental discussion.
>
> **6.Reason for chooing Lyapunov-Razumikhin functions (Q2).** We choose Lyapunov-Razumikhin functions because they rely on an ordinary Lyapunov function on the state space, which is more compatible with our learning and formal verification framework. In contrast, Lyapunov-Krasovskii methods require history-dependent functionals, which are much harder to parameterize with standard neural networks and more difficult to verify formally.
>
> **7.Illustration of reachability analysis (Q3).** We will add a figure to illustrate the reachability analysis in Eqs. (16) and (17) in page 4.

---

> > ### Author Rebuttal · Reviewer_1s4A · 2026-04-03
> >
> > I believe most of my comments are answered in a reasonable way, and I am willing to increase my score.

---

> > > ### Author Response · Authors · 2026-04-03
> > >
> > > Thank you for your time and consideration. We sincerely appreciate the reviewer for raising the score!

---

### Official Review · Reviewer_LKKU · 2026-03-13

**Soundness:** 3
**Presentation:** 2
**Significance:** 3
**Originality:** 2
**Overall Recommendation:** 4
**Confidence:** 4

**Summary:**

This paper proposes a framework for the synthesis and formal verification of neural stability certificates for large-scale interconnected systems subject to communication delays. The core contribution is a sufficient condition for discrete-time scalable input-to-state stability (sISS) utilizing vector Lyapunov-Razumikhin functions (Theorem 4.1). By combining this condition with a two-stage verification strategy and structural isomorphism analysis, the authors claim to achieve $O(1)$ verification scalability relative to the number of agents $N$.

**Compliance With Llm Reviewing Policy:**

Affirmed.

**Final Justification:**

The authors have addressed the primary concern regarding functional decoupling through a detailed sensitivity analysis (Table 1 and Table 2). The data shows that the learned controller indeed utilizes neighbor information significantly, and the performance metrics (Table 3) suggest that the stability certificate does not impose excessive conservatism in the tested scenarios. I decided to change the rating to a Weak Accept.

**Key Questions For Authors:**

1. Could the authors provide the explicit values of the learned coupling gains $\gamma_{ij}$ and the spectral radius $\rho(\Gamma)$ for the experiments in Tables 2-4? This is necessary to verify if the systems studied are actually interconnected or effectively isolated.


2. Can the framework synthesize a verified controller for a system that requires strong coupling to function (e.g., a collaborative manipulation task)? Or does the small-gain condition inherently make such tasks unlearnable? Please give an explicit example.

**Limitations:**

The manuscript fails to provide a transparent and critical self-assessment of its limitations, particularly regarding the trade-off between mathematical verifiability and physical coordination.

1. The authors discuss conservatism almost exclusively in the context of the verification tool (e.g., reachability approximations). By doing so, they avoid addressing the much more severe conservatism of the stability criterion itself. Theorem 4.1 relies on a strictly contractive small-gain condition. This is a very strong decoupling requirement. But there is no discussion on how this constraint prevents the controller from learning complex, high-performance coordination strategies that require agents to be highly sensitive to their neighbors' states.

2. The paper promotes $O(1)$ verification runtime as a main contribution. However, this scalability is a direct byproduct of the strong decoupling requirement. The authors do not discuss the framework's limitations in strongly coupled tasks (e.g., collaborative manipulation or dense swarm obstacle avoidance).

**Strengths And Weaknesses:**

Strengths:

1. Theorem 4.1 provides a formal link between Razumikhin logic (for delay handling) and small-gain conditions (for interconnected stability) in a discrete-time setting.

2. The use of structural equivalence to reuse certificates across agents significantly reduces the verification burden for highly symmetric topologies.

3. Unlike many reachability-based tools that offer finite-time guarantees, this approach provides infinite-time stability certificates.

Weaknesses:

1. The primary concern is that the scalability breakthrough appears to be a direct consequence of the extreme conservatism in Theorem 4.1. To satisfy the small-gain condition in Eq. (4), the coupling gains $\gamma_{i,j}$ must be sufficiently small to be dominated by local damping. This essentially restricts the framework to systems that are "interconnected" only in name, but effectively decoupled in physical reality. The paper lacks an honest discussion on whether this "strong decoupling" requirement prevents the controller from performing any meaningful high-performance coordination task.

2. Following the first weakness, the $O(1)$ scalability is sold as a major technical achievement. However, it seems to be a trivial result of the chosen decoupling strategy. If one assumes a system is so weakly coupled that it satisfies Theorem 4.1, then verifying it agent-by-agent is a natural consequence. The interconnected complexity is bypassed by a mathematical assumption rather than solved by a real algorithm.

---

> ### Author Rebuttal · Authors · 2026-03-31
>
> We first thank the reviewer for the comments and for recognizing the theoretical contributions of this paper. However, we believe there are some misunderstandings regarding both the role of the small-gain condition and the source of the scalability, and we therefore clarify these two points separately below.
>
> **1.Clarification on small gain condition (W1,Q1,Q2,L1).** We respectfully clarify that the small-gain condition in our paper **does not mean that the underlying physical system is weakly coupled**. The reviewer may have mixed up the small gain condition and the weak coupling theorem (see Theorem 2 of [1]), and therefore have mistakenly confused the small-gain coefficients with the physical coupling strength. In our manuscript, **we clearly state that the small-gain condition is imposed on the Lyapunov decremental condition among subsystems (see Eq. (7))**, rather than directly on the system dynamics. In other words, the quantities in the gain matrix characterize how the Lyapunov function of one subsystem enters the decrease inequality of another subsystem, which are analysis-level bounds, not raw physical coupling coefficients. Therefore, requiring the gain matrix to satisfy a small-gain condition does not imply that the system is nearly decoupled. Moreover, in our framework this condition is enforced by construction through the parameterization of the coupling matrix $\Gamma$ (See the details immediately above Eq. (9)).
>
> It is worth noting that **small-gain condition is often introduced precisely to address strongly coupled systems**. For example, [2] applies the small-gain condition to analyse a system with strong couplings between a tracking subsystem and a balancing subsystem, and proves stability of the overall closed-loop system. As stated in its abstract:
> > "Thirdly, to tackle the strong coupling between the tracking and the balancing systems, the small-gain technique is applied for the first time to prove the asymptotic stability of the closed-loop bicycle system."
>
> Moreover, **the microgrid example considered in our paper is itself a strongly coupled interconnected system**. A similar point is also noted in [3]. As stated in the fourth paragraph of the introduction:
> >"...however, a fully decentralized approach may be infeasible due to strong interactions among coupled microgrids."
>
> More broadly, **the use of small-gain condition is a standard practice in the analysis of interconnected systems**. A large body of literature uses small-gain theorem to certify stability of interconnected systems (see, e.g., [2,4,5]). Therefore, adopting a small-gain condition in our paper should not be interpreted as assuming away the interconnection complexity. Rather, it places our approach within a standard compositional stability framework for interconnected nonlinear systems.
>
> [1] Silva, G. F., Donaire, A., Middleton, R., McFadyen, A., & Ford, J. (2024). Scalable input-to-state stability of nonlinear interconnected systems. IEEE Transactions on Automatic Control, 70(3), 1824-1834.
>
> [2] Cui, L., Wang, S., Zhang, Z., & Jiang, Z. P. (2021). Asymptotic trajectory tracking of autonomous bicycles via backstepping and optimal control. IEEE Control Systems Letters, 6, 1292-1297.
>
> [3] Zhang, Y., Xie, L., & Ding, Q. (2015). Interactive control of coupled microgrids for guaranteed system-wide small signal stability. IEEE transactions on smart grid, 7(2), 1088-1096.
>
> [4] Liu, S. J., Zhang, J. F., & Jiang, Z. P. (2007). Decentralized adaptive output-feedback stabilization for large-scale stochastic nonlinear systems. Automatica, 43(2), 238-251.
>
> [5] Lyu, Z., Xu, X., & Hong, Y. (2022). Small-gain theorem for safety verification of interconnected systems. Automatica, 139, 110178.
>
> **2.Clarification on scalability (W2,L2).** First, our scalability does not come from Theorem 4.1 itself, nor from any assumption that directly simplifies the interconnected dynamics. Theorem 4.1 is a certificate condition for the original interconnected system, where the small-gain condition is imposed at the Lyapunov level, and does not change the system topology or physical coupling strength.
>
> Instead, the scalability comes from two concrete verification strategies developed in the paper:
>
> (1) We introduce a reachability-aware two-stage verification procedure, which avoids directly verifying over the full enlarged delayed state space.
>
> (2) We further exploit structural reuse over isomorphic subgraphs, so that equivalent local certificate conditions do not need to be verified repeatedly across the network.
>
> Second, we clarify that our framework does not claim an $O(1)$ complexity, as such asymptotic characterizations are inappropriate for our certificate-based verification architecture. As shown in Theorems 5.10 and 5.12, the actual computational reduction is highly instance-specific, depending on the concrete network topology. Since these factors vary across different cases, a universal asymptotic bound would be technically imprecise.

---

> > ### Author Rebuttal · Reviewer_LKKU · 2026-04-03
> >
> > Thanks for the responses.
> >
> > The authors clarify that the small-gain condition (Eq. 7) is an "analysis-level" bound on Lyapunov functions rather than a direct limit on physical dynamics. However, the physical implication of enforcing this condition "by construction" is that the local control gains must be tuned low enough to ensure that the local Lyapunov decay always dominates all agent interactions. In a truly "strongly coupled" system, such a requirement effectively forces the controller into a decoupled or near-decoupled regime to satisfy the mathematical certificate. Please provide quantitative evidence of a certified policy that maintains strong inter-agent influence without a significant drop in control performance.

---

> > > ### Author Response · Authors · 2026-04-04
> > >
> > > We thank the reviewer for the response. We **have indeed provided an example [1] in the previous rebuttal** to clarify that small gain condition on Lyapunov functions does not necessarily suggest weak coupling. As this concern remains, **we provide an additional example**. We believe these two examples have addressed the concern, and we are happy to further clarify if needed.
> > >
> > > **Additional Example**. The microgrid system considered in the experiments (see the figure in https://anonymous.4open.science/r/Figures_microgrid-2F22/microgrid.png and the system dynamics Eqs. (92)-(96) in the paper) is strongly coupled, as also acknowledged in [2]. This setup is practically meaningful, since many real-world interconnected systems have low-degree interaction structures, where each agent only interacts with a limited number of neighbors. Other examples include platoons, traffic networks, water distribution networks, etc. To further demonstrate that the certified closed-loop system and controller preserve this strong coupling property with enhanced control performance, we conduct a sensitivity analysis to directly quantify how neighboring agents influence a given agent under the learned controller. Specifically, due to symmetry, we consider interver $2$, whose neighbors are intervers $1$ and $3$.
> > >
> > > Table 1 shows the sensitivity results of the closed-loop system when perturbing $\delta_i, i\in\lbrace 1,2,3\rbrace$ of different inverters. We focus on perturbing $\delta_i$ because, according to Eq. (92) in the paper, inter-agent influence is primarily propagated through $\delta_i$. As shown in Table 1, perturbing the local state $\delta_2$ produces the largest effect on all three target states of inverter 2, namely $\delta_2$, $\omega_{\mathrm{error},2}$, and $\xi_2$, which we refer to as the local perturbation effect. However, perturbations in neighboring inverters still lead to noticeable changes, and the system is not decoupled (as stated by the reviewer). In particular, perturbing $\delta_3$ causes changes that are about $28\%$, $68\%$, and $44\%$ of the corresponding local perturbation effect on $\delta_2$, $\omega_{\mathrm{error},2}$, and $\xi_2$, respectively. **This shows that the closed-loop dynamics remain strongly affected by neighboring states.**
> > >
> > > *Table 1: Sensitivity analysis for the closed-loop system*
> > > |Target inverter|Target state|Perturbed inverter|Perturbed inverter initial state|Max change|
> > > |---|---|---|---|---:|
> > > |2|$\delta_2$|1|$\delta_1$|0.103|
> > > |2|$\delta_2$|2|$\delta_2$|1.132|
> > > |2|$\delta_2$|3|$\delta_3$|0.318|
> > > |2|$\omega_{\mathrm{error},2}$|1|$\delta_1$|0.343|
> > > |2|$\omega_{\mathrm{error},2}$|2|$\delta_2$|0.700|
> > > |2|$\omega_{\mathrm{error},2}$|3|$\delta_3$|0.473|
> > > |2|$\xi_2$|1|$\delta_1$|0.171|
> > > |2|$\xi_2$|2|$\delta_2$|0.670|
> > > |2|$\xi_2$|3|$\delta_3$|0.295|
> > >
> > > Table 2 shows the sensitivity of the controller output of inverter 2 to perturbations in different neighboring inverter states. As shown in Table 2, the controller output $u_2$ is also clearly influenced by neighboring states. For example, perturbing $\omega_{\mathrm{error},1}$ causes a larger change in $u_2$ than perturbing the local state $\omega_{\mathrm{error},2}$, and perturbing $\xi_3$ still produces a noticeable effect relative to perturbing $\xi_2$. **This indicates that the controller remains strongly affected by neighboring states, rather than behaving as nearly decoupled subsystems.**
> > >
> > > *Table 2: Sensitivity analysis for the controller*
> > > | Target inverter | Perturbed inverter | Perturbed inverter initial state | Max change |
> > > |---|---|---|---:|
> > > |2|1|$\delta_1$|0.202|
> > > |2|2|$\delta_2$|0.226|
> > > |2|3|$\delta_3$|0.068|
> > > |2|1|$\omega_{\mathrm{error},1}$|1.036|
> > > |2|2|$\omega_{\mathrm{error},2}$|0.245|
> > > |2|3|$\omega_{\mathrm{error},3}$|0.408|
> > > |2|1|$\xi_1$|0.153|
> > > |2|2|$\xi_2$|2.849|
> > > |2|3|$\xi_3$|0.689|
> > >
> > > Table 3 further shows that **the proposed method also improves control performance**, achieving the lowest RMSE under both disturbance levels.
> > >
> > > *Table 3: Control performance*
> > > |Method|0.5 rad/s|1 rad/s|
> > > |---|---:|---:|
> > > |Predictor Feedback|0.082|0.138|
> > > |Nominal Controller|0.082|0.137|
> > > |Compositional ISS|0.079|0.142|
> > > |sISS|0.119|0.171|
> > > |**Proposed Method**|**0.072**|**0.133**|
> > >
> > > **Therefore, even though the certified system satisfies the small-gain condition ($\Gamma_{21}=0.16,\Gamma_{22}=0.34,\Gamma_{23}=0.20$, and $\sum_ {j\in\mathcal{E}_ 2\cup\lbrace 2\rbrace}\Gamma_ {2j}=0.70<1$), the physical coupling between agents remains strong for both the closed-loop system and the controller. Meanwhile, the proposed method does not sacrifice control performance for certification, but instead achieves improved performance over the baselines.**
> > >
> > > [1] Cui et al. (2021). Asymptotic trajectory tracking of autonomous bicycles via backstepping and optimal control. IEEE Control Systems Letters, 6, 1292-1297.
> > >
> > > [2] Zhang, et al. (2015). Interactive control of coupled microgrids for guaranteed system-wide small signal stability. IEEE transactions on smart grid, 7(2), 1088-1096.

---

### Decision · Program_Chairs · 2026-04-30

**Decision:**

Accept (regular)

**Comment:**

The reviewers appreciate the technical novelty and contributions. They recommend improved experiments and discussions on scalability.